# Time-of-day effects of cancer drugs revealed by high-throughput deep phenotyping

Carolin Ector[1,2], Christoph Schmal[3], Jeff Didier [4], Sébastien De Landtsheer [4], Anna-Marie Finger [5,9], Francesca Müller-Marquardt[1,10], Johannes H. Schulte [6,11], Thomas Sauter [4], Ulrich Keilholz[1,7], Hanspeter Herzel[3,8], Achim Kramer [5] & Adrián E. Granada [1,7] ✉

The circadian clock, a fundamental biological regulator, governs essential cellular processes in health and disease. Circadian-based therapeutic strategies are increasingly gaining recognition as promising avenues. Aligning drug administration with the circadian rhythm can enhance treatment efficacy and minimize side effects. Yet, uncovering the optimal treatment timings remains challenging, limiting their widespread adoption. In this work, we introduce a high-throughput approach integrating live-imaging and data analysis techniques to deep-phenotype cancer cell models, evaluating their circadian rhythms, growth, and drug responses. We devise a streamlined process for profiling drug sensitivities across different times of the day, identifying optimal treatment windows and responsive cell types and drug combinations. Finally, we implement multiple computational tools to uncover cellular and genetic factors shaping time-of-day drug sensitivity. Our versatile approach is adaptable to various biological models, facilitating its broad application and relevance. Ultimately, this research leverages circadian rhythms to optimize anti-cancer drug treatments, promising improved outcomes and transformative treatment strategies.

The circadian clock is a central regulator of multiple physiological and behavioral processes found in cyanobacteria, plants, fungi, and animals. In mammals, the hierarchical organization of the circadian system ensures coordinated biological rhythms from the level of the individual cell to the whole organism level[1]. Primate and mouse studies showed that protein-coding genes are rhythmically expressed in a tissue-specific by up to 80% and 40%, respectively[2,3]. These clock-controlled genes regulate key biological processes such as metabolism[4,5], cell proliferation[6], immune response[7], DNA repair, and apoptosis[8].

Disruption of the circadian system is classified as a carcinogen and is associated with multiple cancer subtypes[9–12]. Cancer hallmarks such as sustained proliferation and metastasis[13,14] have been linked to the circadian clock[15,16] and patients with mutations in circadian clock genes exhibit lower survival rates[17–19]. Beyond its role in cancer development, the circadian clock directly interacts with therapeutic targets that affect drug responses[19–21]. Consistent with these observations, recent works have shown that administration of chemotherapeutic agents aligned with the circadian rhythm changes their degree of efficacy

[1]Charité Comprehensive Cancer Center, Charité – Universitätsmedizin Berlin, Berlin, Germany. [2]Faculty of Life Sciences, Humboldt-Universität zu Berlin, Berlin, Germany. [3]Institute for Theoretical Biology, Humboldt-Universität zu Berlin, Berlin, Germany. [4]Department of Life Sciences and Medicine, University of Luxembourg, Esch-sur-Alzette, Luxembourg. [5]Institute for Medical Immunology, Charité – Universitätsmedizin Berlin, Berlin, Germany. [6]Department of Pediatric Oncology, Hematology and Stem Cell Transplantation, Charité – Universitätsmedizin Berlin, Berlin, Germany. [7]German Cancer Consortium (DKTK), Berlin, Germany. [8]Charité – Universitätsmedizin Berlin, Berlin, Germany. [9]Present address: Department of Anatomy, University of California, San Francisco, San Francisco, CA, USA. [10]Present address: Institute of Research for Development, University of Montpellier, Montpellier, France. [11]Present address: Clinic for Pediatrics and Adolescent Medicine, Universitätsklinikum Tübingen, Tübingen, Germany. ✉e-mail: adrian.granada@charite.de

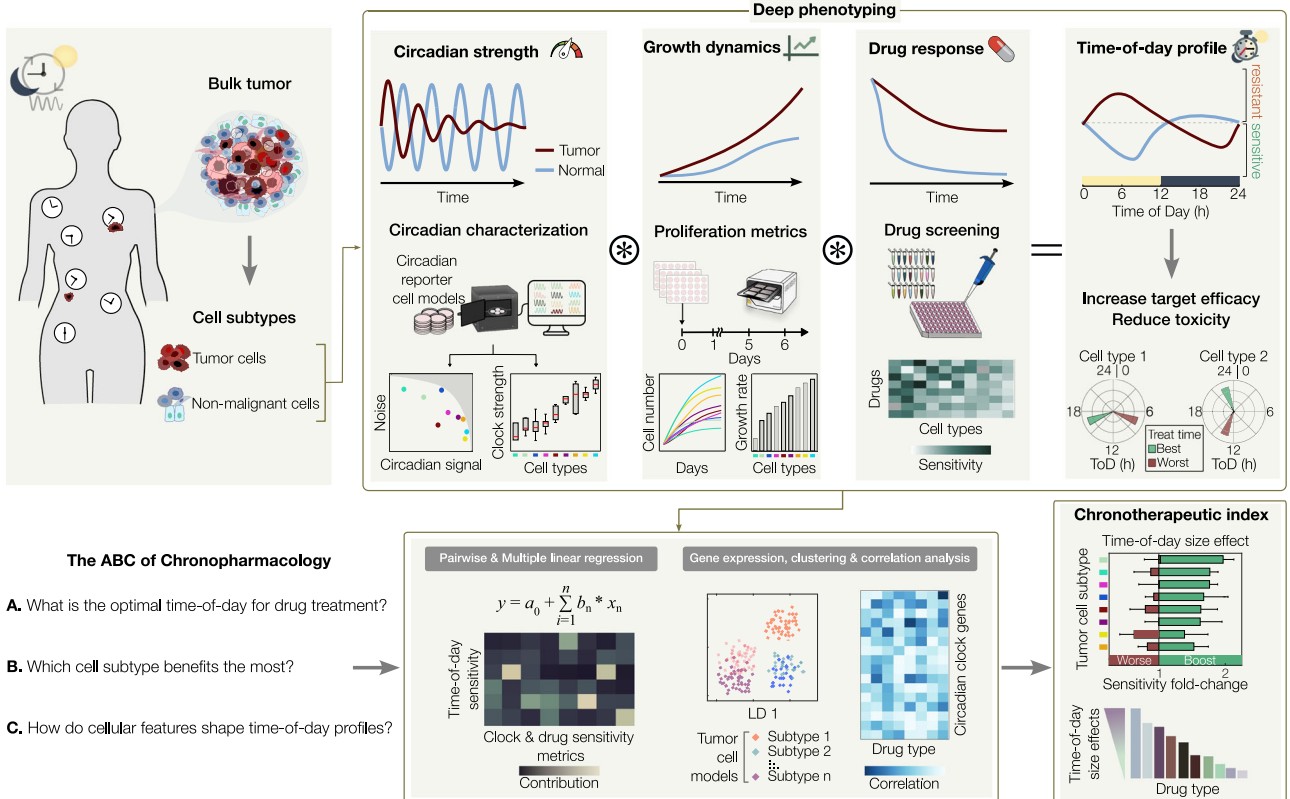

**Fig. 1 | Framework for identifying optimal treatment times in cancer and healthy tissue models.** Schematic of the experimental and computational framework to thoroughly characterize time-of-day drug responses in a variety of cell subtypes, such as cancer and non-malignant cell models. A combination of live recordings is implemented for the deep phenotyping of circadian strength, growth dynamics, and drug responses that shape time-of-day profiles. Using a novel streamlined experimental approach, time-of-day sensitivity profiles are obtained in tumor and non-malignant cell models, providing best and worst timings for increased efficacy and reduced toxicity (top panel). A tandem computational pipeline integrates the deep phenotyping metrics as well as gene expression data of circadian clock genes to quantitatively address three fundamental questions in chronopharmacology (bottom panel). Combining multiple signatures, we define a chronotherapeutic index, ranking cellular models and drug agents by their size-effect gains from drug treatments aligned with the circadian clock (bottom right panel).

throughout the day[20,22–24]. Despite the broad recognition of the benefits of circadian-based drug treatments[22,25,26], an efficient strategy to identify optimal treatment times remains elusive, creating a bottleneck in the implementation. In addition, the mechanisms shaping time-of-day (ToD) sensitivity profiles remain widely unknown.

Here, we introduce a method for the thorough characterization of time-of-day responses in tumor and healthy tissue cell models (Fig. 1). Using an array of experimental and data analysis methods we perform a deep-phenotyping of the critical cellular factors underlying time-of-day responses, i.e., the circadian clock strength, growth dynamics and drug response features. We then deploy a high-throughput strategy to obtain ToD profiles in a panel of drugs and cell line models. Comparing tumor versus non-tumor ToD profiles provides candidate treatment timings to increase efficacy and reduce toxicity. We subsequently integrate our dataset with publicly available gene-expression databases to rigorously address three fundamental questions in the field of circadian pharmacology, known as chronopharmacology. These questions are: (A) What is the optimal time of day for drug treatment? (B) Which cell subtype benefits the most from circadian-aligned drug treatment? and (C) How do cellular features shape time-of-day profiles? Finally, we define a chronotherapeutic index, ranking cellular models and drug agents that stand to gain the most benefit from circadian-based treatments.

## Results

### Deep circadian phenotyping in cancer cell models

Despite the increasing recognition of the role of the circadian clock in cancer progression and treatment response, the extent to which different cancer subtypes maintain circadian rhythmicity remains poorly understood. To address this and quantitatively characterize the degree of rhythmicity of cancer cell models, we implement an approach that integrates recordings of circadian clock activity with comprehensive time-series analysis techniques, as depicted in Fig. 2a. To robustly characterize the circadian clock molecular network, we monitored the positive and negative feedback arms of the molecular clock using a combination of two circadian luciferase reporters for *Bmal1* and *Per2* (Fig. 2b). In Fig. 2c we show representative raw luciferase signals from the breast cancer cell line MDAMB468 and the corresponding detrended and normalized signals[27]. As expected for a robust functional clock network, *Bmal1* and *Per2* signals show stable anti-phasic expression patterns throughout the recording (Fig. 2c and Supplementary Fig. 1a, b).

### Circadian clock strength varies in cancer and healthy cell models

Following the acquisition of *Bmal1* and *Per2* expression dynamics, we conducted a comprehensive circadian phenotyping of the signals. To accurately capture different aspects of the circadian clock dynamics, we implemented a strategy that integrates three complementary time-series analysis techniques, i.e., autocorrelation (AC), continuous wavelet transform (CWT), and multiresolution analysis (MRA). The rationale behind using this complementary analysis is to harness the strengths of each technique; AC for identifying stable temporal features, CWT for revealing time-dependent amplitude and period changes, and MRA for extracting multi-scale features, ensuring a comprehensive understanding of the signal dynamics. Using this approach we screened a broad panel of cell models, including non-malignant breast epithelial MCF10A, cancer cell models of various

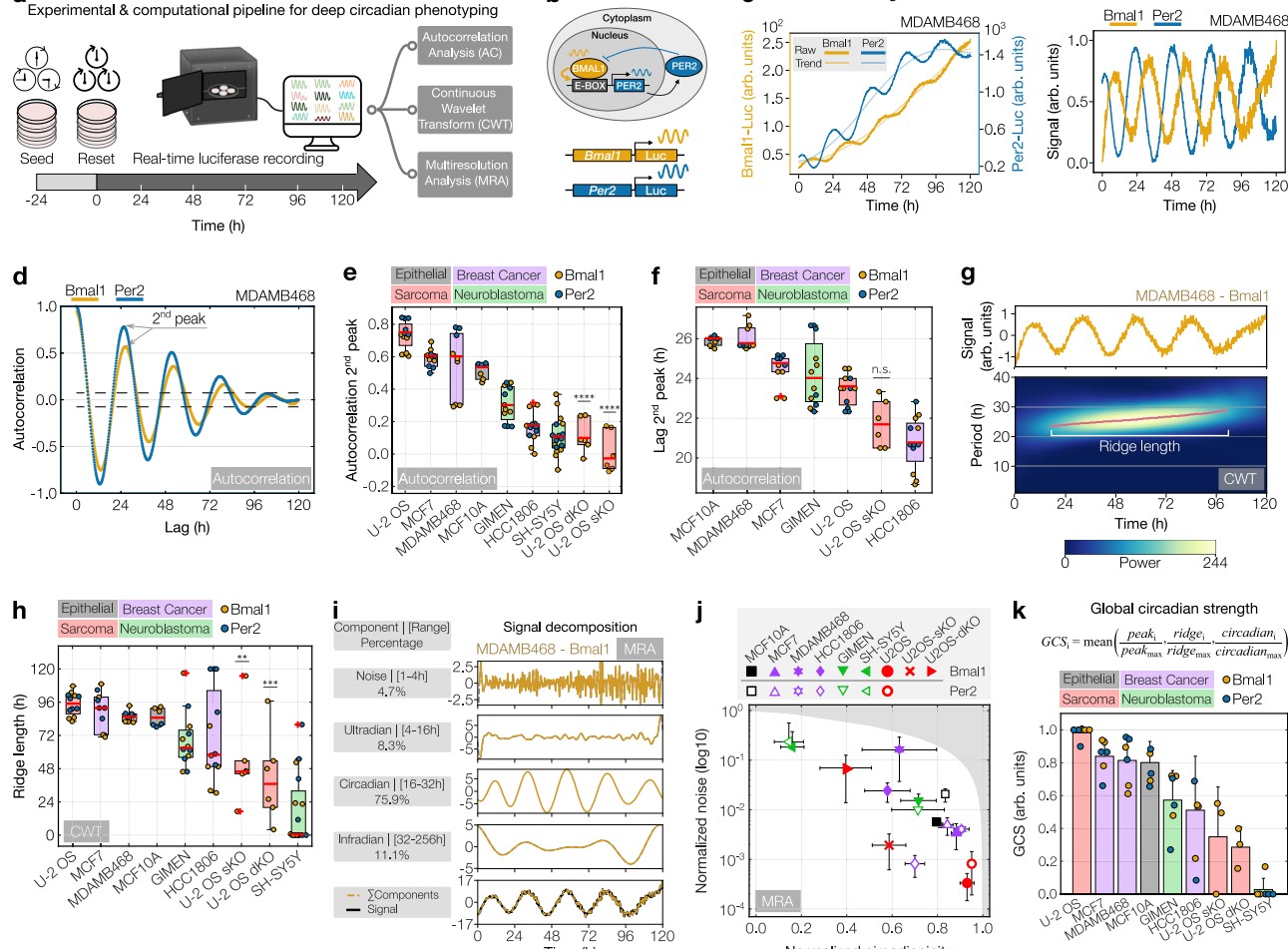

**Fig. 2 | Determining circadian clock strength in cancer and healthy tissue cell models. a** Schematic of the deep circadian phenotyping approach. **b** Simplified circadian feedback loops involving *Bmal1* and *Per2*. **c** Raw and processed signals from MDAMB468-*Bmal1/Per2*-Luc cells. **d** Example of autocorrelation (AC) analysis. The arrow indicates 2nd peak and abscissa (lag). Dashed lines = 95% CI. **e** Boxplot of AC values and (**f**) lags of signals from various cell models. **g** Wavelet power spectrum (bottom) from continuous wavelet transform (CWT) showing time-resolved periods of detrended-amplitude-normalized signals from MDAMB468-*Bmal1*-Luc cells (top). Red line = main oscillatory component (ridge). **h** Boxplot of CWT ridge lengths from various cell models. Box bounds in (**e**, **f**, **h**) are defined by the 25th and 75th percentiles. Extending whiskers represent data points within 1.5 times the interquartile range from lower and upper quartiles. Red lines and crosses denote the median and outliers, respectively. *n* = 12 samples, collected from *Bmal1*-/*Per2*-reporters, with 6 samples per reporter (*n* = 2 biological replicates á technical triplicates or duplicates [HCC1806 *Per2*-Luc]). *n* = 6 for U-2 OS KO-lines (*Bmal1*-Luc-only) and MCF10A (single experiment). *n* = 9 for MDAMB468 (*Per2*-Luc: single

experiment). *n* = 17 for SH-SY5Y (biological triplicates with technical triplicates or duplicates). **i** Multiresolution analysis (MRA) of detrended MDAMB468-*Bmal1*-Luc signal. % = fraction to signal. **j** Scatterplot of normalized MRA noise versus circadianicity components from the indicated cell models. The shaded area covers an unattainable range. Data represents the mean ± s.d. of multiple samples per reporter cell line (see above). **k** Bar diagram ranking cell models by global circadian strength, integrating min-max scaled parameters from AC (peak), CWT (ridge), and MRA (circadianicity) for *Bmal1*-Luc and, where applicable, *Per2*-Luc signals. Data represents mean ± s.d of scaled parameters (*n* = 6, except U-2 OS knockouts where *n* = 3 parameters). Only the positive s.d. is shown. Color coding in (**b–l**, **k**) corresponds to *Bmal1* (yellow) and *Per2* (blue) reporters. Color coding of cell models in (**e**, **f**, **h**, **j**, **k**) corresponds to tissue origin. One-way ANOVA and Tukey's post-hoc test compared U-2 OS WT and KO cell lines, where **, ***, and **** indicate *p*-values of $5.7 \times 10^{-3}$, $4.8 \times 10^{-4}$ and ≤ 0.0001, respectively. n.s. = non-significant. Source data for (**c–k**) are provided as a Source Data file.

entities (breast cancer, neuroblastoma, and sarcoma) as well as two knockout variants of osteosarcoma U-2 OS cells with a single deletion in the circadian clock gene *Cry1* (U-2 OS sKO) or paired with a deletion in the *Cry2* locus[28] (U-2 OS dKO).

Autocorrelation is a robust method for estimating the periodic quality of a signal, particularly for time series whose properties remain stable over time, known as stationary signals. Calculating the autocorrelation function from each recording provides the strength and period values, obtained from the second peak ordinate and abscissa (lag), respectively (Fig. 2d). This analysis showed a wide range of circadian strengths across models with the U-2 OS wild-type ranking highest with a median autocorrelation value of 0.74, whereas U-2 OS sKO and U-2 OS dKO variants ranked lowest with median correlation

values of − 0.04 ($p = 7.7 \times 10^{-12}$) and 0.09 ($p = 1.4 \times 10^{-10}$), respectively (Fig. 2e). Heterogeneity between and within cancer types was further observed for the oscillation period ranging from short periods of ˜ 21 h in HCC1806 and U-2 OS sKO cells ($p = 0.83$) to longer periods of ˜ 26 h in MDAMB468 and MCF10A cells. U-2 OS dKO (34.8 h, $p = 8.2 \times 10^{-5}$) and the neuroblastoma cell line SH-SY5Y (39.2 h) showed periods well above the circadian range and were excluded from this analysis (Fig. 2f). Consistent with these results, detrended signal traces of the different cell models and additional breast cancer cell lines indicate high variability in circadian clock signals across the models tested (Supplementary Fig. 1c).

To capture non-stationary features of circadian signals, such as unstable periods and fluctuating amplitudes, we implemented

continuous wavelet transform, a prevalent technique for analyzing dynamic temporal signals[29]. Figure 2g shows a CWT power spectrum heatmap of the *Bmal1* signal from the MDAMB468 example. The heatmap displays signal components (0–40 h period) over the 120-h recording, color-coded by relative power (see "Methods"). Time-connected regions of high relative power mark the signal's main oscillatory component, referred to as the ridge. Strong signals exhibit continuous long ridges whereas weaker signals have short and discontinuous ridges. As a complementary measure of clock strength, we quantified the ridge length from all our recordings. Our analysis shows that U-2 OS cells have a well-maintained clock with a median ridge length of 4.1 days, closely followed by MCF7, MDAMB468, and MCF10A cells (3.4–3.8 days). U-2 OS sKO and dKO cells showed ridges shorter than two days (1.9 days [$p = 5.7 \times 10^{-3}$] and 1.5 days [$p = 4.8 \times 10^{-4}$], respectively) (Fig. 2h).

While both autocorrelation and continuous wavelet transform, provide insights into the most significant signal component, they do not quantify how the signal is distributed among non-circadian frequencies. To obtain more comprehensive signal information and an analogous signal-to-noise metric, we next implemented multi-resolution analysis. MRA involves decomposing the detrended signal into four distinct component bands, namely the noise (1–4 h), ultradian (4–16 h), circadian (16–32 h), and infradian (32–48 h) components (see "Methods"). In Fig. 2i, we present an example of MDAMB468 *Bmal1*-Luc where 75.9% of the signal is in the circadian range, 4.7% in the noise and the remaining 19.4% in the ultradian or infradian range. To obtain an analogous signal-to-noise measure of all recorded cell models, we plotted the circadian component ("circadianicity") versus the noise component and observed a broad range of ratios (Fig. 2j). Consistent with our previous analysis, U-2 OS cells show strong circadian signal with the lowest noise levels (<1%) and the highest proportion of circadian components (~ 93%) for both reporters. Knockout of *Cry1* or *Cry1/Cry2* reduced the circadian component to 59% ($p = 5.8 \times 10^{-6}$) and 40% ($p = 2.0 \times 10^{-8}$), respectively, while increasing the noise component by 5.8-fold in the single knockout and by 209-fold in the double knockout. This signal-to-noise analysis map indicates that signals from the *Per2*-reporter were slightly more circadian and less noisy than those from the *Bmal1*-reporter (Fig. 2j and Supplementary Fig. 1d). *Bmal1* and *Per2* exhibit unique oscillatory patterns, reflecting distinct biological pathways within the circadian clock system. Their individual behaviors might offer valuable insights into the functionality of the circadian rhythm. To streamline our analysis and facilitate comparison, we averaged the values of *Bmal1* and *Per2* in Fig. 2e, f, h, and k. However, for a more detailed examination, separate analyses for each can be found in Supplementary Fig. 1e, f.

Finally, to obtain a global strength metric, we normalized each circadian parameter to its respective maximum value across all tested cell line models and computed the mean, facilitating a gradual ranking of cell models from low to high circadian strength (Fig. 2k, see "Methods").

In summary, by using a multi-faceted approach to characterize circadian rhythms in the different models tested, we identified heterogeneous circadian clock phenotypes, suggesting a strong clock in U-2 OS, MCF7, MDAMB468, and MCF10A cells and an impaired but present clock in GIMEN and HCC1806 cells. These results challenge the common expectation that most cancer cells have a weak clock and underscore the significance of defining gradual metrics of circadian strength in a model-specific manner.

### Growth and drug response dynamics in cancer cell models

Together with circadian potency, cell growth dynamics and how cells respond to drug treatment in time are expected to influence responses throughout the day. Thus, we next evaluated growth characteristics and drug sensitivities across a spectrum of drug and cell line models. To showcase our approach's ability to detect within-subtype differences, we examined nine cell lines of the triple-negative breast cancer (TNBC) subtype alongside the non-malignant MCF10A breast cell model. The TNBC cell lines were analyzed across several molecular subtypes as classified by Lehmann et al., specifically the basal-like 1 (BL1), basal-like 2 (BL2), and mesenchymal-like (MES) TNBC subtypes. Growth and drug sensitivity assays in cancer models often rely on single time point measurements (e.g., ATP-based viability and replating assays) that provide a time-averaged snapshot. However, cancer cells respond dynamically to drugs, with temporally evolving effects, so conclusions drawn from single time point data can potentially mask the true effects of drugs. To accurately capture the dynamics of growth and drug response, we implemented a time-resolved live-cell imaging setup, directly counting cell nuclei in a fluorescence channel, while simultaneously evaluating confluency in the complementary brightfield channel (Fig. 3a). Figure 3b shows representative snapshots of both imaging channels from MDAMB468 cells under untreated conditions. Frame-by-frame quantification provides growth trajectories for each cell model tested (Supplementary Fig. 2a), with MDAMB468 cells showing a 5.3-fold increase in growth and reaching a ~100% confluency after 4 days (Fig. 3c). Quantification of doubling times (DTs) calculated from confluency and cell numbers correlated well across the 10 cell line models tested ($R^2 = 0.69$) (Supplementary Fig. 2b). Upon examining the normalized cell number trajectories, we observed substantial variability in growth across the cell line models, ranging from approximately 2- to 16-fold over the 4-day recordings (Fig. 3d). To capture growth signatures from the entire recording, we fitted an exponential function to each time series and obtained the growth rate *k* for each trajectory (Fig. 3e). As expected, cell number growth rates and doubling times exhibit a strong anticorrelation ($R^2 = 0.83$) (Fig. 3f and Supplementary Fig. 2c).

Next, we studied the response of the cancer cell models upon treatment with a panel of seven drugs targeting a broad range of mechanisms and pathways, i.e., the DNA synthesis inhibitors 5-FU and doxorubicin, mitosis-inhibiting alisertib and paclitaxel, the PI3K/AKT/mTOR inhibitor torin2, as well as cisplatin and olaparib which target the DNA damage response (DDR) pathway (Fig. 3g). We then tracked responses to a broad range of drug concentrations, as exemplified in Fig. 3h by MDAMB468's olaparib treatment. Here, doses up to 1.5 μM resulted in slightly slower growth relative to the control, while 6.3 μM of olaparib significantly hindered growth, and ≥25 μM led to total inhibition and cell death. Fitting an exponential function to growth curves yielded positive growth rate values for weak doses while higher doses resulted in negative growth rates, indicating population decline. Using this approach, we next quantified the response and stratified all tested cell lines and drugs.

### Multi-parametric evaluation of drug sensitivity reveals heterogeneity between and within models

Accurate assessment of drug effects is essential for identifying exploitable weaknesses in cancer treatments. Traditional drug sensitivity metrics, like the IC$_{50}$ (the drug concentration that reduces cell counts by 50% relative to the control), can be greatly influenced by factors such as the assay duration and the number of cell divisions between drug administration and the final evaluation of drug sensitivity. These factors can inadvertently introduce artefactual correlations and lead to misinterpretations of drug sensitivity results. To obtain robust drug sensitivity metrics, we employed the normalized growth rate inhibition (GR) approach as described by Hafner et al.[30]. Here, growth rates under treated and untreated conditions are compared and normalized to a single cell division. Fitting a dose-response equation to GR-values yields five drug sensitivity parameters, namely, the concentrations at half-maximal effect (GEC$_{50}$) and at which GR = 0.5 (GR$_{50}$), the drug effect at the infinite concentration (GR$_{inf}$), the steepness of the sigmoidal fit (*Hill coefficient*), and the area over the curve (GR$_{AOC}$) (Fig. 3i). Moreover, the GR value directly reflects cellular response phenotypes, indicating

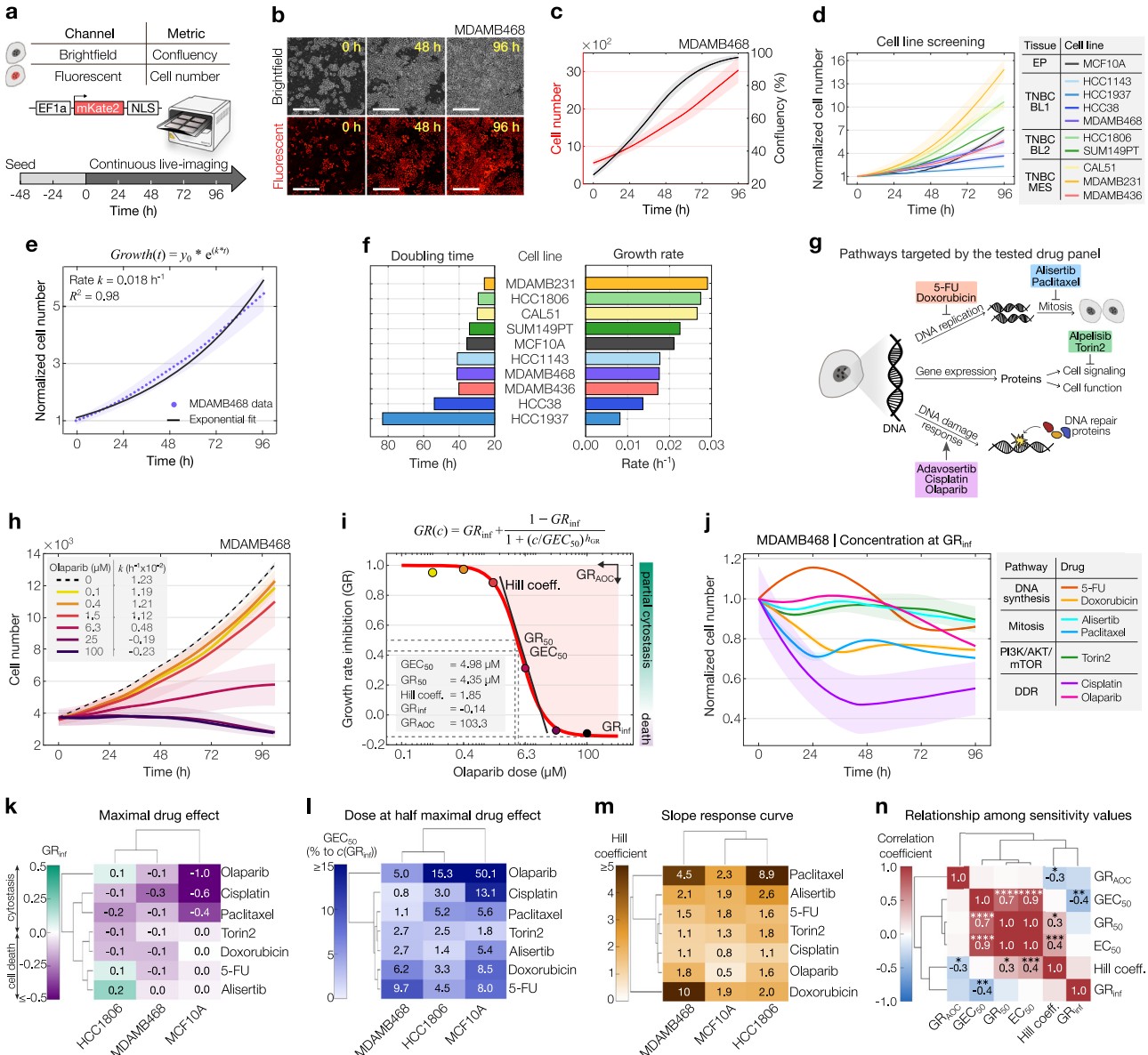

**Fig. 3 | Unraveling growth and drug response dynamics through long-term live-cell imaging. a** Schematic of the experimental setup. NLS = nuclear localization sequence. **b** Snapshots of MDAMB468 growth in brightfield (top) and red-fluorescent channel (bottom). Ruler = 400 μM. **c** MDAMB468 confluency (black) and cell numbers (red) over time. **d** Normalized growth curves of indicated cell models. BL1, BL2, and MES refer to TNBC subtypes basal-like-1/-2, and mesenchymal-like, respectively. EP = epithelial. **e** Exponential fit (solid line) for MDAMB468 growth curves, yielding growth rate ($k$), and fit accuracy ($R^2$). Dots represent normalized cell numbers averaged across 9 images taken per well. The shaded area represents the standard deviation. **f** Bar diagrams of doubling times and growth rates, sorted in descending order from highest to lowest growth rates. Parameters calculated from growth curves, averaged across 9 images taken per well (CAL51, HCC38, HCC1806, HCC1937, MDAMB231, MDAMB468), or across six control wells from later described time-of-day experiments. **g** Schematic of pathways targeted by drugs used in this study. **h** Cell numbers and growth rates of MDAMB468 cells treated with varying olaparib doses (color-coded) or solvent (dashed line). Data

represents the mean±s.d. of two plates. **i** Dose-response curve of $GR$-values, highlighting various drug sensitivity metrics. The underlying data corresponds to the example shown in (**h**). Error bars=95% CI. **j** Normalized cell numbers of MDAMB468 treated with approximate $GR_{inf}$ doses of the indicated drugs. Data represents the mean ± s.d. of two plates or mean±s.e.m of 9 images taken on a single plate (cis-platin). **k–m** Hierarchical clustering of drug sensitivity parameters across cell-drug combinations. $GEC_{50}$-values are shown relative to the approximate $GR_{inf}$ dose. **n** Pearson's correlation coefficients of sensitivity parameters shown in (**k–m**) and additional combinations (Supplementary Data 1; $n = 50$ cell-drug combinations per parameter, except for $EC_{50}$-values where $n = 49$ due to fitting constraints). Denoted are significant pairwise correlation coefficients (two-sided test with no adjustments made), where * indicate $p$-values of 0.03 (Hill coeff. vs. $GR_{AOC}$) or 0.018 (Hill coeff. vs. $GR_{50}$), **$p$-value of 0.0016, ***$p$-value of 0.0015, and ****$p$-values ≤ 0.0001. Data in (**k–n**) is based on the mean of two plates, or 9 images taken per well on a single plate (cisplatin). Source data for (**b–f**, **h–n**) are provided as a Source Data file.

partial growth inhibition for values between 0 and 1, complete cytostasis for a value of 0, and signifying cell death in the range from 0 to −1. Interestingly, when the same cell line was treated with different cancer drugs at their respective concentration evoking a saturating $GR_{inf}$ response, distinct drug-dependent trajectories emerged (Fig. 3j). This underscores the importance of a time-resolved approach for

characterizing drug sensitivity across different drugs and, accordingly, across distinct cell line models.

We next compared a subset of drug-sensitivity metrics and cell line models for the different drugs tested (Fig. 3k–m). While most drugs were able to induce death in the subset of cell lines to varying degrees, we identified three drugs that resulted in partial growth

inhibition of HCC1806, namely olaparib, 5-FU and alisertib (Fig. 3k). Hierarchical clustering of $GR_{inf}$ −values revealed a grouping of the two TNBC cell lines and while this clustering was maintained for the $GEC_{50}$-values (Fig. 3l), the Hill coefficient resulted in a different clustering of the cell lines (Fig. 3m). To explore the potential relationships between various sensitivity metrics, we combined data from multiple drugs and cell models and computed cross-correlations, revealing significant associations among $GEC_{50}$, $GR_{50}$, and $EC_{50}$-values ($p \leq 0.0001$) (Fig. 3n). In addition, significant yet less pronounced correlations emerged between the Hill coefficient and different drug sensitivity parameters, whereas the $GR_{inf}$ and $GR_{AOC}$-values showed minimal correlation with other metrics. For complete $GR$ curves and information on additionally tested cell lines and drugs, see Supplementary Fig. 3 and Supplementary Data 1.

## Variability throughout the day depends on the drug and cell model

The presence of a robust circadian clock combined with drugs that efficiently affect growth in a tumor model are promising prerequisites for identifying drug variability throughout the day. However, these prerequisites do not provide a priori insights into the specific time-of-day response profile. To systematically screen ToD drug sensitivities, we developed an experimental strategy designed to significantly reduce the investigator's workload and number of consecutive drug perturbations, thereby increasing throughputness, reproducibility and accuracy (Fig. 4a). In this approach, cell populations are seeded 24 h prior to the start of continuous live-cell imaging followed by performing a 3-step circadian clock resetting protocol in which separate cell populations receive a dexamethasone pulse at three different times (0, 4, 8 h). To test cells in later stages of the circadian cycle, drugs are administered at their estimated half-effective dose 32 and 48 h after the first resetting step, creating a range of time differences between reset and treatment of 0, 4, 8, 16, 20 and 24 h in relative circadian time (Fig. 4a, *right panel*). The effects of the different treatment times on cell growth were monitored by live-cell imaging up to day 6, enabling the evaluation of drug responses for 4 days in both treatment groups, as shown for the alisertib-treated TNBC cell line HCC1937 in Fig. 4b. In typical ToD assays, cells are constantly proliferating during the 24 h of ToD treatments, resulting in varying cellular densities at the time of the drug treatments. These varying cell densities have the potential to influence drug responses in vitro, possibly concealing or introducing bias when determining ToD-specific drug effects. To account for different cell densities at the time of treatment, drug responses are determined as the ratio of the number of cells at the time of each treatment to the number of cells 96 h after each treatment, keeping the time window from treatment to evaluation identical across conditions (Fig. 4c, left panel). To highlight relative response differences within a day, results are presented relative to the circadian time of 0 h. Values > 1 indicate increased resistance, while values < 1 indicate higher sensitivity compared to treatment at time 0 h (Fig. 4c, right panel). We quantify the maximum variability in relative responses as the ToD Maximum Range ($ToD_{MR}$).

Our next objective was to explore the variability of ToD profiles of different drugs within a subset of TNBC cells. To accomplish this, we screened ten cell line models shown in Fig. 3 treated with eight different drugs, generating approximately 80 ToD profiles (Fig. 4d, e and Supplementary Fig. 4a, b). Comparing examples from individual cell lines, namely HCC1937 and MCF10A, revealed that HCC1937 exhibited relatively conserved ToD profiles, whereas the non-malignant cell line showed greater ToD variability for the different drugs (Fig. 4d). Furthermore, ToD sensitivity profiles varied significantly from drug to drug (Fig. 4e and Supplementary Fig. 4b),

suggesting that ToD profiles may be highly dependent on the cancer cell model and drug mechanism of action.

To assess varying time-of-day sensitivities across all tested cell line models and drugs, we calculated the corresponding $ToD_{MR}$ value for each drug-cell combination (Fig. 4f). Averaging the $ToD_{MR}$-values per drug and cell line revealed a gradual ranking of ToD variability for each drug and cell model tested (Fig. 4g). Within the drug panel, the highest and lowest ToD variability was observed for cisplatin and alisertib, respectively, with a ~2-fold difference in the average $ToD_{MR}$ value (Fig. 4g, top panel). Similarly, the ToD variability of the tested cell lines varied by ~2-fold, with MCF10A showing the highest and HCC38 the lowest average variability. Considering only cancer cell lines, SUM149PT ranked highest and showed a similar degree of ToD sensitivity variability as the non-malignant cell line (Fig. 4g, bottom panel).

To assess the impact of circadian clock disruptions on ToD-dependent drug sensitivity, we evaluated circadian-perturbed U-2 OS *Cry1/2*-dKO cells, alongside wild-type cells that demonstrated the strongest circadian rhythms in our assessments (see Fig. 2). Using our ToD treatment approach (see Fig. 4a), we tested three drugs which elicited high ToD response variations in our breast cancer panel (see Fig. 4g), and which target distinct molecular pathways. Our results demonstrate substantial ToD-dependent drug sensitivity in WT cells and markedly reduced responses in *Cry1/2*-dKO cells (Supplementary Fig. 4c). Specifically, $ToD_{MR}$-values decreased by 62% for cisplatin, 58% for paclitaxel, and 40% for 5-FU in *Cry1/2*-dKO compared to WT cells (Supplementary Fig. 4d), highlighting a critical role of the circadian clock in influencing drug sensitivity throughout the day.

Together, our findings reveal distinct ToD profiles across most tested drugs and models, highlighting individualized ToD sensitivity within distinct drugs and cell models despite the common TNBC categorization and shared drug-target pathways.

## Determining treatment times for maximum drug effect

The critical factor in the design of future chronotherapeutic treatments may be not only the sensitivity of cancer cells but rather the differential time-of-day sensitivity between cancer and non-malignant tissues[26]. To illustrate this, we compared the ToD response profiles of TNBC cancer cell models with the profile of MCF10A, as shown for HCC1937 and alisertib in Fig. 4h. In this case, the cancer and non-malignant cell models show an antiphasic ToD response profile, with an almost inverted profile. By calculating the greatest response differences between the two cell models, we determined the treatment times of maximum and minimum benefit (Fig. 4h–j). Analyzing the entire panel of tested drugs per cell line model and plotting polar density histograms of the times of maximum and minimum benefit, we found that the highest overall benefit is achieved at 10–12 h and 18–20 h in the treatment of HCC1937, while earlier treatment times of the day yield minimum benefit. In contrast, SUM149PT and MDAMB231 show a single prominent time window throughout the day that provides maximum treatment benefit (Fig. 4i). In addition to cell model-to-model variability in the benefit times, we explored the drug-to-drug variability by calculating the polar density histogram for the individual drugs tested across all cell line models (Fig. 4j). 5-FU showed a clear preference for administration between 8 and 10 h, while torin2 and paclitaxel showed more variability in maximum and minimum benefit times, indicating different response relationships to the non-malignant cell model across the cancer cell lines tested.

Finally, we quantified the extent of maximum and minimum treatment benefit by calculating the corresponding fold changes in ToD responses between the cancer and non-malignant cell model (Fig. 4k). By averaging these fold changes across cell lines and drugs, we established a ranking referred to here as the "chronotherapeutic index". While the average ToD variability highlights the benefits of a single model (Fig. 4g), the chronotherapeutic index reveals distinctions between the cancer models and the non-malignant MCF10A cell

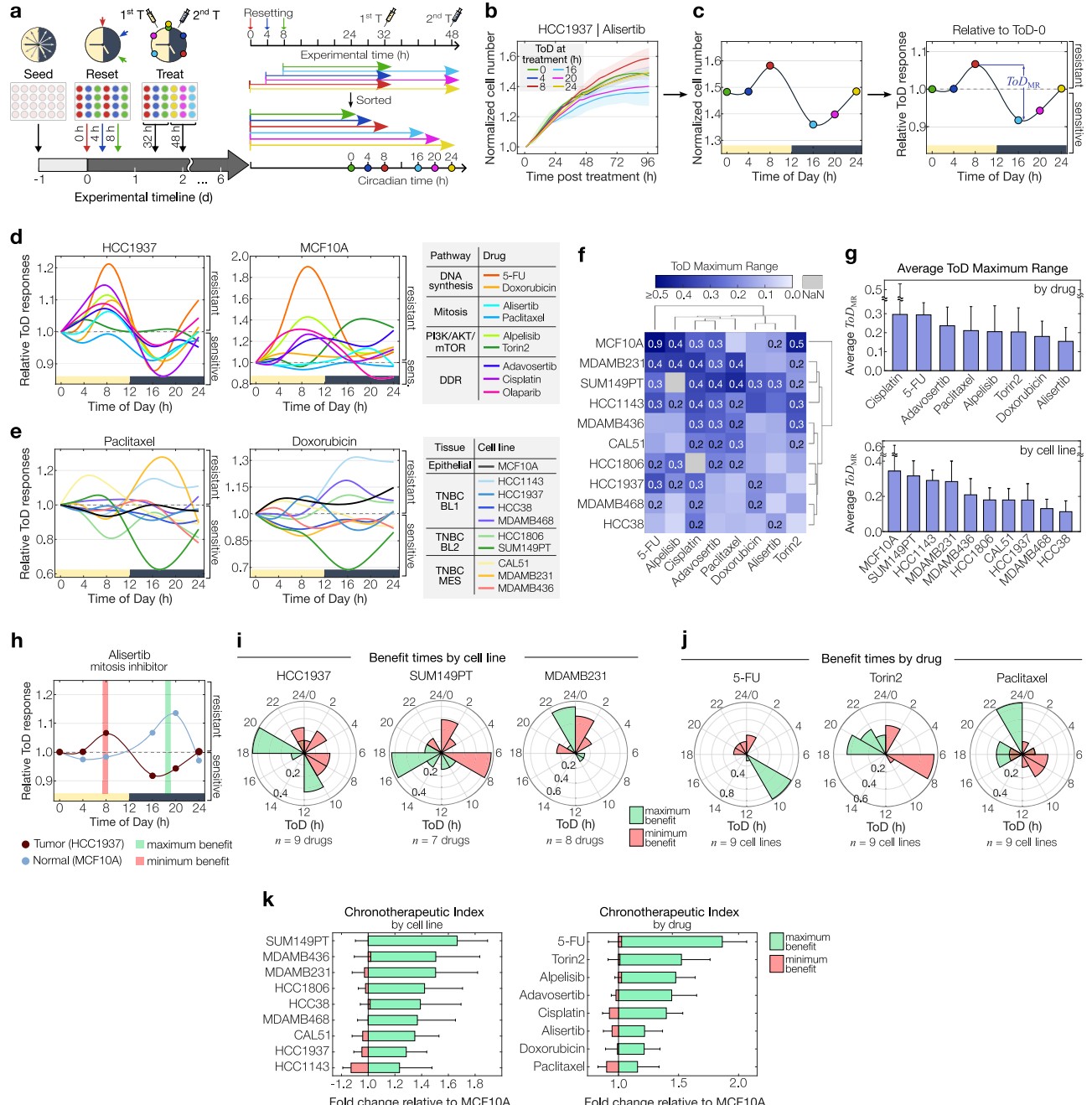

**Fig. 4 | Time-of-day drug sensitivity is drug and tissue model dependent.**
**a** Schematic of the experimental setup to screen for time-of-day (ToD) responses. Three clock-resetting steps are performed in 4-h intervals. Drugs are administered 32 or 48 hours post initial reset, resulting in six circadian times (0, 4, 8, 16, 20, 24 h). Growth is monitored by long-term live-cell imaging. Timelines on the right depict experimental (top) and relative circadian times (bottom), color-coded by each ToD tested. **b** Cell counts of HCC1937 treated with alisertib at different times of the day. Normalization to the respective time of treatment. **c** Cell numbers 4 days post-treatment versus treatment times, corresponding to (**b**) (left). ToD response curve (ToD-RC) depicting relative responses to ToD 0 h. Blue arrows mark the maximum ToD response range ($ToD_{MR}$) (right). **d** ToD-RCs for HCC1937 (left) or MCF10A (right) cells treated with different drugs (color-coded). **e** ToD-RCs for 10 cell models treated with paclitaxel (left) or doxorubicin (right). Color coding of cell models according to tissue origin. **f** Hierarchical clustering of $ToD_{MR}$-values across drug-cell

combinations. Values above 0.2 are shown. Data clustering with the UPGMA method and Euclidian distance. **g** Bar diagrams ranking $ToD_{MR}$-values per drug (top) and cell model (bottom). **h** ToD-RCs for alisertib-treated HCC1937 tumor and MCF10A non-tumor cells overlaid, yielding maximum and minimum benefit times. Data shown in (**b**–**h**) represents the mean (± s.d.) of two plates. **i** Polar histograms for benefit times across cell models (*n* = 7–9 drugs, as indicated in the figure) and **j**, drugs (*n* = 9 cell models). **k** Butterfly charts depicting fold changes relative to MCF10A at benefit times, averaged per cancer cell model (left,) and drug (right). Color-coding of maximum and minimum benefit times is shown in (**h**–**k**) in green and red, respectively. Data are shown in (**g** and **k** represents mean ± s.d. of tested cell models per drug (*n* = 9 cell models; alpelisib and cisplatin: *n* = 8) and vice versa (*n* = 8 drugs; HCC1806 and SUM149PT: *n* = 7). For clarity, one-sided error bars are shown. Source data for (**b**–**k** are provided as a Source Data file.

model, thereby offering valuable insights into the potential advantages of adopting a chronotherapeutic-based schedule. It's worth noting that the rankings based solely on average ToD variability within a single model or drug didn't align completely with the ranking of the chronotherapeutic index. This emphasizes the importance of evaluating effects in relation to healthy tissues when considering chronotherapy applications (Fig. 4g, k).

To sum up, our findings emphasize distinct chronotherapeutic response dynamics in vitro between cancer and healthy cell models, underscoring two key objectives of chronotherapy: pinpointing optimal treatment times to maximize cancer toxicity while minimizing impacts on healthy tissues.

## The relationship between time-of-day profiles and clock, growth, and drug sensitivity metrics

The time-of-day sensitivity in cancer models results from complex interactions involving the circadian clock, cancer cell growth, and drug responses. However, the specific mechanisms governing these interactions remain largely unknown. To address this, we employ statistical tools, including linear regression, dominance analysis, and determinant ranking, to uncover the main parameters influencing a key aspect of ToD sensitivity curves, namely the maximum range in responses ($ToD_{MR}$) (Fig. 5a). To explore how individual metrics relate to $ToD_{MR}$-values, we first performed pairwise linear regression analysis (Fig. 5b–f). We considered each circadian channel individually or combined and found the strongest and most significant correlations for clock metrics from the *Bmal1* channel (Fig. 5d and Supplementary Fig. 5a, see Supplementary Data 2 for a complete set of circadian metrics across the cell models tested). $ToD_{MR}$-values of 5-FU were significantly correlated with amplitudes ($r = 0.8$, $R^2 = 0.6$) and the prominence of the circadian component (circadianicity) of the *Bmal1* signal ($r = 0.7$, $R^2 = 0.43$), whereas other circadian clock metrics were poorly associated (Fig. 5b, d). On the other hand, we observed no significant and rather low associations for $ToD_{MR}$-values and growth metrics as well as for $ToD_{MR}$-values and drug sensitivity metrics (Fig. 5c, e, f). To identify the best associated individual metric and drug, we integrated all correlations and generated a ranking as shown in Fig. 5g. For the different metrics, we observed the highest correlation for the amplitude ($r = 0.43 \pm 0.02$, mean ± s.e.m.). Among the individual drugs, adavosertib ranked highest in overall correlation ($r = 0.39 \pm 0.03$, mean ± s.e.m.) between $ToD_{MR}$-values and the different metrics. Relative average associations between all metrics and $ToD_{MR}$-values were predominantly positive, with only five metrics demonstrating inverse relationships (Supplementary Fig. 5b). Notably, only one drug, paclitaxel, displayed average negative associations across all metrics. To further test the associations between individual cellular parameters and $ToD_{MR}$-values, we applied a linear regression model and compared calculated versus actual $ToD_{MR}$-values of up to five new cell line models in Bland-Altman plots (Supplementary Fig. 5c, see Supplementary Data 3 for the complete new dataset). This revealed mean biases of −0.04, −0.05, and 0.06 between calculated and actual $ToD_{MR}$-values for the clock, growth, and drug sensitivity metrics, respectively. As also shown in Supplementary Fig. 5c, mean biases are near zero and most predicted data points lie within the limits of agreement, which indicates minimal overall bias and good agreement between the predicted and observed data points.

While linear regression approaches are robust for examining individual metrics, they do not provide relative information about which metrics have the most significant impact on $ToD_{MR}$-values. To address this, we used dominance analysis and systematically tested all possible combinations of metrics. By measuring how much each metric improved the accuracy of the model when added or removed, we identified the highest contributing ones to the $ToD_{MR}$-values of the different drugs. For alpelisib, our analysis identified the amplitude of *Bmal1* signals as the most important factor, accounting for 42% of the

observed ToD sensitivity variability (Fig. 5h). In contrast, paclitaxel and adavosertib showed the most homogeneous distribution of individual contributions, with no single metric appearing especially important. Considering the shares of each determinant combined for all drugs led to a ranking of determinants as shown in Fig. 5i. Here, we observed the largest contributions for the *Bmal1* amplitude (mean = 21.9%) and smallest for the Hill coefficient of drug response curves (mean = 9.9%).

In summary, we were able to identify systematic dependencies of the maximal time-of-day drug sensitivities and different circadian clock, growth, and drug sensitivity parameters. We have further shown that the relative importance of the different determinants varies substantially depending on the specific drug, highlighting the importance of considering multiple factors in understanding the time-of-day dependent variability of drug response in a cell model.

## Differential impact of core clock genes in shaping time-of-day sensitivity of cancer models

Beyond a cancer model's circadian clock, growth, and drug features, the expression levels of core circadian clock genes are likely to contribute to the time-of-day sensitivity profiles. Thus, we explored the connection between the expression patterns of 16 essential clock genes and the maximum range in ToD sensitivity profiles ($ToD_{MR}$). We defined core clock genes as those that directly control the transcriptional-translational feedback loops of the molecular clock network and whose dysregulation or mutations result in disrupted circadian rhythms[31,32]. The selected gene panel displayed distinct expression patterns across the TNBC cell lines tested (Supplementary Fig. 6a). To examine potential relationships between core clock genes and $ToD_{MR}$-values, we used three different methods, namely linear correlation analysis, dimensionality reduction by linear discriminant analysis (LDA) and principal component analysis (PCA) (Fig. 6a).

Linear correlation analysis revealed significant correlations between $ToD_{MR}$-values of five tested compounds from our drug panel and selected core clock genes (Fig. 6b). Focusing on the mitosis inhibitor paclitaxel, we identified particularly strong anticorrelations with the expression levels of *Per3* ($r = -0.88$, $R^2 = 0.78$) and *Dbp* ($r = -0.72$, $R^2 = 0.51$) (Fig. 6b, c). Yet, the overall correlation between circadian clock genes and $ToD_{MR}$-values was relatively weak, which was also apparent when accumulating absolute correlations per drug or per metric (Supplementary Fig. 6b). Here, the highest overall correlation of $ToD_{MR}$-values were found for cisplatin ($r = 0.40 \pm 0.21$, mean ± s.d.), while *Per2* was most associated with $ToD_{MR}$-values among the circadian clock gene panel ($r = 0.48 \pm 0.20$, mean ± s.d.) (Supplementary Fig. 6b).

To assess the cumulative impact of circadian gene expression on the strength of time-of-day effects as indicated by the $ToD_{MR}$-values, aiming to uncover more complex patterns that might not be explainable through linear correlations alone, we proceeded to apply LDA on a drug-by-drug basis. To do so, we first categorized cell models into two groups based on the median $ToD_{MR}$-value for each drug. Figure 6d provides an example using paclitaxel, effectively separating the data into groups of high and low $ToD_{MR}$-values. Implementing LDA to examine the individual linear discriminant components of each circadian clock gene revealed a clear ranking of genes in terms of their importance in discriminating the cell lines into the two $ToD_{MR}$ groups. For paclitaxel, the core clock genes *Cry2*, *Dbp*, and *Per3* were the most important contributors to the specific discrimination, collectively accounting for approximately 46% of the discriminative information (Fig. 6d, right panel). Importantly, the contribution of each gene in discriminating between $ToD_{MR}$-response groups varied for the tested drugs, suggesting that either the drug or the drug targets may interact differently with the molecular components of the circadian clock (Supplementary Fig. 6c–i). To explore the circadian clock's impact on ToD sensitivity across various drug treatments, we assessed the overall contribution of each gene by cumulating their effects across all drugs. This revealed modest overall contributions, with the highest ranking

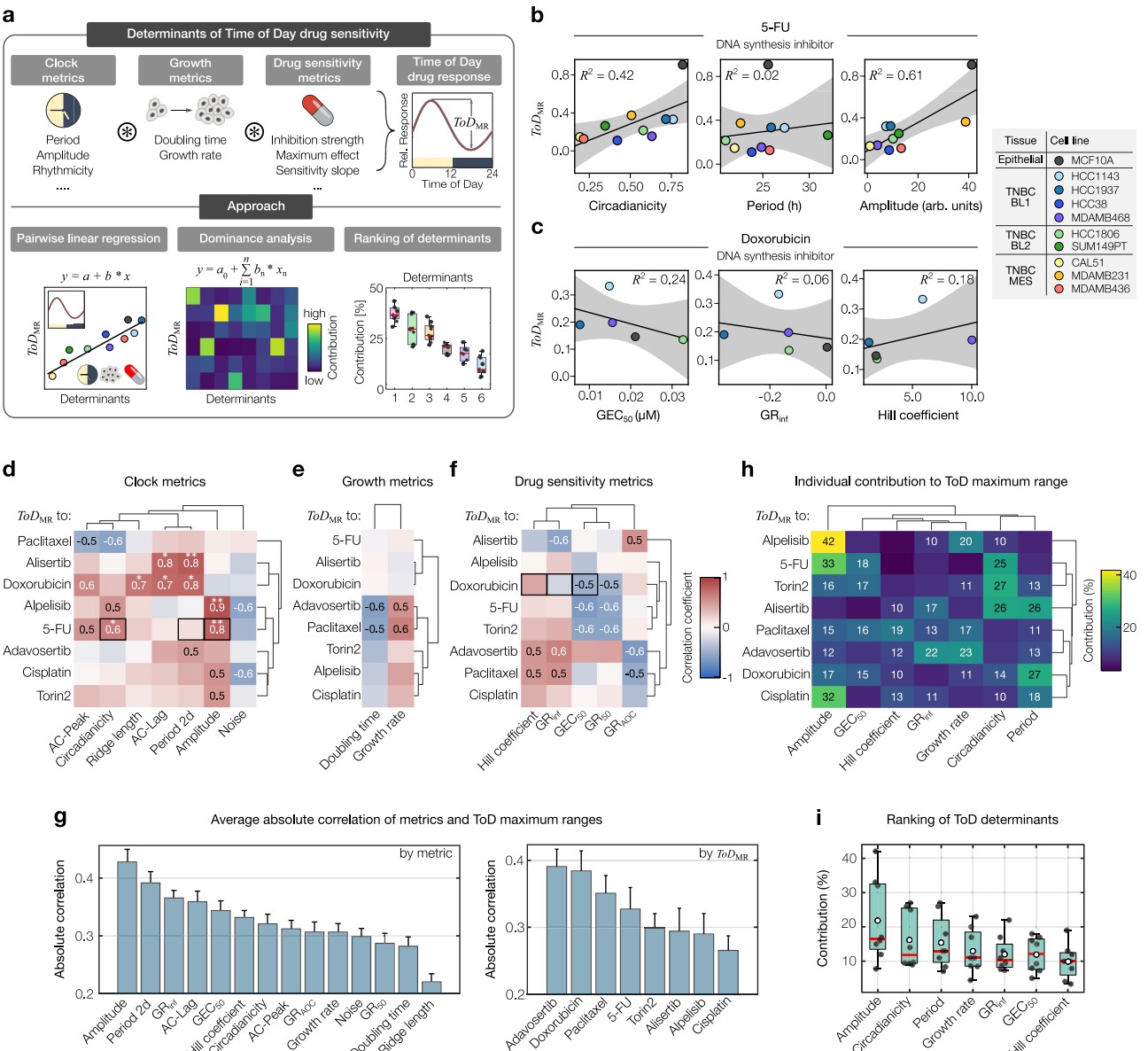

**Fig. 5 | Clock and drug sensitivity metrics shape ToD curves. a** Computational approach to investigate how circadian rhythms, cell growth dynamics, and drug sensitivity factors influence the time-of-day (ToD) drug efficacy. **b** and **c** Example of linear correlations between $ToD_{MR}$ and circadian (**b**) or drug sensitivity parameters (c), for 5-FU ($n = 10$ cell models) and cisplatin ($n = 5$ cell models), respectively. Color coding of cell models according to tissue origin. The central black line represents the regression line. Gray-shaded area = 95% CI of the linear regression fit. Model accuracy is indicated by $R^2$-values. **d**–**f** Hierarchical clustering of Pearson correlation coefficients ($r$) between $ToD_{MR}$-values of different drugs (rows) and the respective metric (columns) for clock strength (**d**, $n = 10$ cell models), growth (**e**, $n = 10$ cell models) or drug sensitivity (**f**, $n = 5$ cell models). $r$-values ≥ 0.5 and ≤ -0.5 are shown. Black rectangles indicate examples shown in (**b** and **c**). Significant pairwise correlations (two-sided test with no adjustments made) are indicated by stars, where *, and **, denote $p$-values ≤ 0.05 and 0.01, respectively. Exact $p$-values: Alisertib-AC-Lag = 0.011; Alisertib-Period-2d = 0.004; Doxorubicin-Ridgelength =

0.030; Doxorubicin-AC-Lag = 0.014; Doxorubicin-Period-2d = 0.012; Alpelisib-Amplitude = 0.002; 5-FU-Circadianicity = 0.044; 5-FU-Amplitude = 0.007. Note: sample size in (**d**–**f**) for alpelisib is $n = 1$; and for cisplatin = 9. **g** Bar diagrams ranking the absolute correlation between $ToD_{MR}$-values, and each metric depicted in (**d**–**f**), ranked by metric ($n = 8$ drugs) or by drug ($n = 14$ metrics). Data represents the mean ± s.e.m. For clarity, one-sided error bars are shown. **h** Hierarchical clustering of the dominance analysis matrix showing the individual contribution of the circadian clock, growth, and drug sensitivity parameters (columns) in predicting drug-dependent $ToD_{MR}$-values (rows). Colors indicate the percentage contribution, as detailed in the color bar. See key for d–f for sample sizes ($n$). **i** Boxplot of the overall contribution of cellular metrics to predict $ToD_{MR}$-values, corresponding to (**h**). Box bounds are defined by the 25th and 75th percentiles. Extending whiskers represent data points within 1.5 times the interquartile range from lower and upper quartiles. Red lines denote the data's median, and white circles the mean. Source data for b–i are provided as a Source Data file.

*Rorβ* accounting for 10.5% ± 9.6% (mean ± s.d.) of the discriminative information (Fig. 6e).

While LDA is a powerful method to determine the discriminatory contribution of the individual core clock genes, PCA allows for identifying underlying patterns and relationships among the different genes. For paclitaxel, PCA distributed the cell models well along the first two principal components which combined explained 73.7% of the

variability among the different ToD sensitivity profiles (Supplementary Fig. 6j, left panel). Ranking of the PC loadings showed that the degree of contribution of each circadian clock gene in shaping the PCA outcome was highly individual. For paclitaxel, *Csnk1d* was the major contributor to the ToD sensitivity variability within the first principal component, while along the second principal component, *Clock* ranked as the main contributor (Supplementary Fig. 6j, right panels).

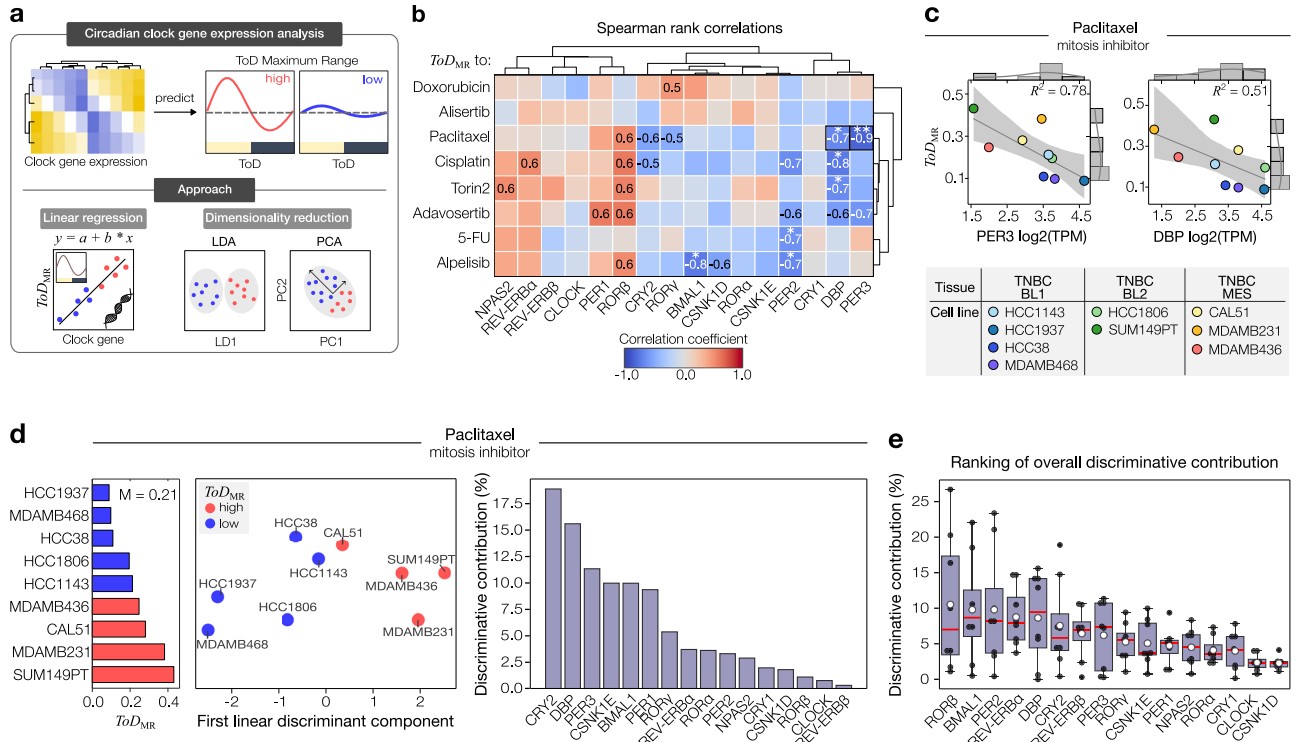

**Fig. 6 | Gene expression analysis unveils diverse roles of circadian genes in determining time-of-day patterns. a** Approach to identify the role of core clock genes in determining the strength of time-of-day (ToD) drug sensitivity. **b** Hierarchical clustering of Spearman rank correlation coefficients between drug-dependent $ToD_{MR}$-values (rows) and core clock genes (columns). Black boxes indicate examples shown in (**c**). Statistical significance of the correlations are indicated as stars, where * and **, denote $p$-values $\leq 0.05$ and $0.01$, respectively. Exact $p$-values: Paclitaxel-$Per3 = 0.002$; Paclitaxel -$Dbp = 0.030$; Cisplatin-$Dbp = 0.021$; Torin2-$Dbp = 0.036$; 5-FU-$Per2 = 0.050$; Alpelisib-$Bmal1 = 0.021$; Alpelisib-$Per2 = 0.047$. Cisplatin and alpelisib: $n = 8$ cell lines, else $n = 9$ cell lines. **c** Example of linear correlation analysis between paclitaxel $ToD_{MR}$-values and $Per3$ or $Dbp$ expression levels measured as log2(TPM). Color-coded data points indicate different cancer cell models. The gray continuous line indicates the normal distribution fit of the samples. Gray-

shaded area = 95% CI of the linear regression fit. Model accuracy is indicated by $R^2$-values. Top and lateral histograms indicate the count of samples along the range of gene expression, and $ToD_{MR}$-values, respectively. $n = 9$ cell lines. **d** Linear discriminant analysis (LDA) on median-based binarized $ToD_{MR}$-values for paclitaxel. Cell models with $ToD_{MR}$-values below or above the median (M) are colored in blue and red, respectively (middle and left panel). The contribution to the obtained discriminative information is shown in percentage for each of the circadian clock genes (right panel). **e** Boxplot showing the mean-based ranking of overall discriminative contributions of each circadian clock gene across all tested drugs. $n = 8$ drugs. Box bounds are defined by the 25th and 75th percentiles. Extending whiskers represent data points within 1.5 times the interquartile range from lower and upper quartiles. Red lines denote the data's median, and white circles the mean. Source data for b–e are provided as a Source Data file.

Collectively, our findings suggest limited associations between individual expression levels of core clock genes and the observed ToD maximum ranges within the examined cell lines. However, we found interesting relationships when considering the circadian clock gene panel as a collective, indicating potential connections between the molecular circadian clock and drug-dependent time-of-day sensitivity.

## Discussion

While the fundamental role of the circadian clock in influencing disease progression and treatment response is widely recognized[22,33,34], our understanding of the underlying biological mechanisms remains fragmented. Approaches that enable the precise study of drug perturbations and treatment responses within controlled circadian-timed environments play a pivotal role in advancing our understanding of the effects regulated by the circadian clock. To date, there has been a notable absence of standardized methodologies for the comprehensive exploration of the chronotherapeutic potential of drugs and cancer models. In this work, we present an integrative approach for the high-throughput quantification of time-of-day response profiles and the elucidation of circadian, growth, and drug sensitivity factors that shape these profiles (Fig. 1). This innovative method holds significant promise for advancing biological discovery, offering a robust platform

to explore the intricate interplay between circadian rhythms, cellular growth, and drug timing efficacy.

Despite the general expectation that the circadian clock is likely dysregulated in highly transformed cancers[19,33], our approach reveals robust rhythms in numerous cancer cell models (Fig. 2). Furthermore, our assessment of circadian rhythms in non-malignant epithelial cells, luminal breast cancer cells, and osteosarcoma cells aligns with prior research[28,35–37], underscoring the versatility of our approach in elucidating circadian clock properties across various cancer and tissue types.

Among these, we focused on triple-negative breast cancer, a highly aggressive and heterogeneous subtype, representing about 15% of diagnosed breast tumors. Current therapies are ineffective, with cytotoxic chemotherapy still being an integral part of treatment despite limited efficacy and severe side effects[38]. Our investigation into the growth dynamics and drug responses of multiple TNBC cell models has generated a diverse array of metrics. Notably, our results indicate that our set of drug sensitivity metrics, namely the $GR_{50}$, $GEC_{50}$, $GR_{AOC}$, $GR_{inf}$, and the Hill coefficient, exhibit milder correlations to each other, which is in agreement with recent studies[39]. This challenges the conventional binary classification of sensitive versus resistant models and suggests that drug sensitivity rankings are rather drug-dependent and sensitivity metric-specific (Fig. 3).

By implementing our screening approach for the profiling of drug sensitivities across different times of the day, we generated approximately 80 distinct time-of-day sensitivity profiles, unveiling a wide range of sensitivity variations across drugs and cell line models (Fig. 4 and Supplementary Fig. 4). We then ranked models by their time-of-day sensitivity and observed up to 30% response differences throughout the day for the top scoring models. We further detected substantial variation in time-of-day sensitivity among the panel of drugs tested, revealing time-of-day dependent drug effects. Consistent with previous findings, we show time-of-day dependent variability in responses to the DNA synthesis inhibitors 5-FU[22,23,40] and doxorubicin[22,41] as well as to the DNA intercalator cisplatin[22,41]. Furthermore, we have introduced a chronotherapeutic index, which gauges the relative advantages of adopting a circadian-based drug schedule. Various non-malignant cell models may be employed for calculating this index, tailored to the particular disease model under investigation. For example, fibroblast models, whose strong circadian rhythms have been described in several works[42–44], are expected to show unique relationships with cancer models in terms of sensitivity patterns at different times of the day.

Delving into the molecular mechanisms that potentially govern time-of-day sensitivity, our method has unraveled key relationships between circadian clock dynamics, growth patterns, and drug sensitivities, all of which significantly impact time-of-day drug responses. It is noteworthy that the specific contribution of these parameters varies depending on the drug under consideration, underscoring the multifactorial nature of time-of-day dependent variability in drug reactions (Fig. 5). Aiming to simplify the circadian analysis and to provide an overarching view of circadian clock dynamics, we initially combined *Bmal1*- and *Per2*-signals in Fig. 2. However, separate evaluations of the circadian genes, as presented in Supplementary Fig. 1e and f, proved crucial to elucidate the significant impact of oscillatory *Bmal1* dynamics on $ToD_{MR}$-values.

In an exploratory approach, we identified limited associations between expression levels of core circadian clock genes and ToD sensitivities (Fig. 6). This may hint at a more complex role of the molecular circadian clock network in this context, where distinct associations might be masked by high redundancy and interconnection between core clock genes. In addition, our results could indicate a discrete role of the cell cycle in shaping our observed ToD sensitivity profiles[45–47], which also invites for further research in that context.

Assessing the cumulative impact of circadian gene expression on $ToD_{MR}$-values, we found differential contributions of each circadian gene in discriminating between high and low ToD-dependent sensitivity based on the drug tested. To investigate whether this observation underlies variations in the expression levels of the drug targets, follow-up explorations involving e.g., mRNA or protein quantification techniques performed at different times of the day could be conducted in future studies focusing on the molecular mechanisms of ToD sensitivity.

Given their recognized influence on pharmacodynamics[22,33,34], our study primarily focused on circadian rhythms. Nevertheless, our multiresolution analysis of *Bmal1* and *Per2* oscillatory signals uncovered distinct alternate rhythms of shorter and longer periods, as illustrated in Fig. 2i, j. The significant impact of independent 12-h ultradian rhythms on stress responses has been previously documented for mammalian cells[48] and constitutes an interesting field for future research in the scope of our time-of-day drug sensitivity profiles.

Despite employing a robust method to introduce luciferase reporters and validating several circadian results, as well as using population-based recordings of non-clonal cell lines, our approach does not rule out the potential impact of random reporter insertions on circadian clock estimation. Moreover, while our findings are primarily based on in vitro models and are not directly applicable to treatment recommendations, our framework can be directly applied to model organisms of higher complexity, such as 3D patient-derived organoids and animal models where the circadian clock can be tracked.

Altogether, we provide a comprehensive method to determine optimal drug treatment times for complex diseases and to identify the cell subtypes that will most benefit from a time-of-day based treatment. Leveraging innovative experimental and computational techniques our approach further elucidates the specific biological features that shape time-of-day response profiles, thereby advancing our understanding of the cellular mechanisms that drive time-of-day sensitivity. As a result, our method provides tools to uncover the cellular mechanisms that govern time-of-day sensitivity, paving the way for new biological discoveries.

## Methods
### Experimental methods
**Cell culture.** HCC1143, HCC1806, HCC1937, HCC38, and MDAMB468 cells were obtained from the American Type Culture Collection. BT549, CAL51, MDAMB231, MDAMB436, and SUM149PT cells were kindly provided by the Sorger lab (Harvard Medical School, Ludwig Cancer Center, Boston, USA). MCF10A and MCF7 cells were kindly gifted by the Brugge lab (Harvard Medical School, Ludwig Cancer Center, Boston, USA). GIMEN and SH-SY5Y cells were provided by the Schulte lab (Universitätsklinikum Tübingen, Clinic for Pediatrics and Adolescent Medicine, Thübingen, Germany) and the U-2 OS reporter cell lines by the Kramer lab (Charité, Institute for Medical Immunology, Berlin, Germany). MCF10A cells were maintained according to the Brugge lab's media recipe based on DMEM/F12 media (Gibco, 11320033) supplemented with 5% horse serum (Gibco, 26050088), 20 ng/ml EGF (Peprotech, AF-100-15), 0.5 mg/ml Hydrocortisone (Sigma, 50-23-7), 100 ng/ml Cholera Toxin (Sigma, 9012-63-9), 10 µg/ml Insulin (Sigma, I1882) and 1% penicillin-streptomycin (Pen-Strep, Gibco, 15140122). All other cell lines were maintained in RPMI 1640 (Gibco, 11875-093) supplemented with 10% fetal bovine serum (FBS) (Gibco, A5256701) and 1% Pen-Strep. For bioluminescence recordings and long-term imaging, cells were cultured in FluoroBrite DMEM medium (Gibco, A1896701) supplemented with 10% FBS, 300 mg L-Glutamine (Gibco, 25030024) and 1% Pen-Strep. Cells were kept at 37 °C in a humidified 5% $CO_2$ environment and regularly tested for mycoplasma.

**Generation of reporter cell lines.** To produce lentivirus carrying constitutive red-fluorescent nuclear reporters, HEK293T cells at 80% confluence were transfected with a mix of 1.8 µg gag/pol packaging plasmid (Addgene #14887), 0.7 µg pRev packaging plasmid (Addgene #12253), 0.3 µg VSV-G envelope plasmid (Addgene #14888) and 3.2 µg of a plasmid with a EF1α-mKate2-NLS sequence. Lentivirus expressing either *Bmal1*- or *Per2*-promoter driven luciferase reporter was produced by transfecting HEK293T cells with 6 µg psPAX2 (Addgene #12260), 3.6 µg pMD2G (Addgene #12259) and 8.4 µg lentiviral expression plasmid (pAB-mBmal1:Luc-Puro or plenti6-mPer2:Luc-Blast, respectively). Transfections were conducted using Lipofectamine 3000 (Invitrogen, L3000015) according to the manufacturer's instructions, and media was replaced by RMPI 1640 medium supplemented with 10 mM HEPES (Gibco, 15630080) before adding the transfection mixture. Lentiviral supernatant was collected after 48 h and 72 h and passed through a 0.45 µm filter (Millipore, HAWP04700). For lentiviral transduction, receiver cells at 70% confluence were incubated with a mix of 1 ml lentivirus-containing supernatant, 8 µg/ml protamine sulfate (Sigma, P4020), and 10 µM HEPES for 6 h. Ensuing, cells were washed with PBS (Gibco, 10010-015) and cultured in their regular culture medium for 2 days before starting antibiotic selection of transduced cells. To select for transduced cells, cells were grown in a medium containing 5 µg/ml blasticidin (Adooq, A21608) or 2 µg/ml puromycin (Gibco, A1113803) according to the resistance cassette on the lentiviral expression plasmid until non-transduced control cells died. Details about the generation of

the U-2 OS *Cry1*-sKO, U-2 OS *Cry2*-sKO, and U-2 OS *Cry1/Cry2*-dKO cell lines can be found in the original publication by Börding et al.[28].

**Bioluminescence recordings.** Cells expressing *Bmal1*- or *Per2*-promoter driven luciferase reporters were seeded in 35-mm dishes (Nunc) to reach approximate confluence on the following day. To account for potentially different phases of single-cell circadian clocks in cell populations, we collectively reset the circadian clocks by administering a standard dose of 1 μM dexamethasone[49,50] (Sigma, D4902, dissolved in EtOH). After 30 min incubation, cells were washed once with PBS, and an imaging medium supplemented with 250 μM D-Luciferin (Abmole, M9053) was added. To avoid evaporation of the media throughout the bioluminescence recordings, the dishes were sealed with parafilm as described in Finger et al.[49]. Luminescence intensity was monitored at 10-min intervals for 5 days in an incubator-embedded luminometer (LumiCycle, Actimetrics). Bioluminescence recordings were conducted using two biological replicates. For each individual experiment, three, in some cases only two, technical replicates were assessed. MCF10A cells have been tested on a single day using three technical replicates.

**Long-term live-cell imaging.** All live-cell imaging experiments were performed with cells expressing the fluorescent mKate2-NLS nuclear reporter, seeded in 48-well plates (Falcon) at a density that saturates at unperturbed growth conditions towards the end of each experiment. Long-term live-cell imaging was conducted using an incubator-embedded Incucyte live-cell widefield microscope (Essen BioScience). Cells were imaged in the brightfield and for nuclei segmentation and cell counting in the red channel (excitation: 567–607 nm, emission: 622–704 nm). Images were analyzed by frame-by-frame nuclei counting with the in-built Incucyte software and results were further processed in MATLAB. For experiments that involved drug treatments, cells were seeded 1 day before starting the live recordings, and images were taken using a 4x magnification lens in 2 fields-of-views per well every 1–2 h for a total duration of 4–6 days. Two independent plates per condition were assayed in a single experiment. For drug response experiments with cisplatin, a single plate per condition was assayed in a single experiment, imaged in 9 fields-of-views per well using a 10x magnification lens. For experiments capturing unperturbed growth dynamics, 300–1000 cells, corresponding to ~10% confluency, were seeded 2 days before starting the live recordings. Using a 10x magnification lens, 9 images per well were acquired in 1–2 h intervals for a total duration of 4 days.

**Dose-response curves.** Drug stock solutions (100–10 mM) were prepared in DMSO and preserved at −20 °C. Cisplatin (Sigma, 232120) stock solution of 3.33 mM was prepared in 0.9% NaCl and stored at room temperature. For dose-response assays, a serial 5–6 point log4 dilution of each drug was freshly prepared in its solvent prior to treatment. Following concentration ranges were tested: 100–0.1 μM for 5-fluorouracil (5-FU, Sigma, 03738-100MG), alpelisib (Biozol, TGM-T1921-10MG) and olaparib (Adooq, A10111); 10–0.01 μM for torin2 (Sigma, SML1224-5MG) and alisertib (Hölzel, S1133-5); 10–0.04 μM for adavosertib (Biocat, T2077-5mg-TM); 1–0.004 μM for doxorubicin (Hölzel, A14403-100); and 0.4–0.0004 μM for paclitaxel (Hölzel, M1970-50mg). Cisplatin was tested in a serial 10-point log2 dilution with doses ranging from 70–0.14 μM. Compounds were added to the cells one day after cell seeding in a drug-media mixture of 9% of the total well volume. Solvent-only treated control cells were assayed along the treated conditions. Each tested condition contained equal amounts of the solvent. Cell growth was monitored by live-cell imaging for at least 4 days as described above.

**Time-of-day treatments.** Cells were seeded and allowed to attach overnight. The next day, live recordings commenced as described

earlier. We performed independent resetting steps every 4 h over an 8-h period, creating distinct cell populations at 0, 4, or 8 h of circadian time. Subsequently, we administered the same drug concentration, which corresponded to the estimated half-maximal effective concentration, at either 32 or 48 hours after the initial resetting step. This allowed us to simultaneously test six different circadian stages (0, 4, 8, 16, 20, and 24 h). Drug concentrations were determined separately for each cell line and drug combination. For cisplatin, the treatments we performed were slightly different, using 4 independent resetting steps every 3 h over a 9-h period followed by drug treatments 32 or 48 h post initial resetting. In all cases, cell growth was continuously monitored through long-term live-cell imaging for a total of 6 days.

## Computational methods
### Time-series analysis of circadian signals
**Detrending.** Raw time-series data were detrended by applying a sinc filter with a 48-h cut-off period using the open-source software package pyBOAT[27] (v0.9.1) within the Anaconda Navigator (v1.10.0).

**Amplitude envelope and normalization.** Continuous amplitude envelope calculation was obtained using continuous wavelet transform implemented in pyBOAT with a time window of 48 h. The amplitude normalization was done by taking the inverse of the envelope of the detrended signal as described in Mönke G, et al.[27].

**Autocorrelation analysis.** Periodicity data was assessed by calculating its autocorrelation and abscissa at the second peak using the 'autocorr' and 'findpeaks' MATLAB functions[51] from the detrended time series.

**Continuous wavelet transform.** The main oscillatory component, known as the ridge component, was obtained using a wavelet-based spectral analysis from amplitude-normalized and detrended signals. For the ridge detection, we used both an adaptable threshold and a fixed threshold. The adaptable threshold was based on each signal's half-maximal spectral power and was implemented for the "period" and "phase difference" metrics shown in Fig. 2 and Fig. S1. The instantaneous phase difference between the *Bmal1* and *Per2* signals was calculated using the MATLAB 'atan2' function (Supplementary Fig. 1a) and for the polar histogram representation, we deployed the 'polarhistogram' function and the 'Circular Statistics Toolbox' (v1.21.9.0. by Philipp Behrens) from MATLAB (Supplementary Fig. 1b). For the analysis of "amplitudes" and "ridge lengths" shown in Fig. 2 and the determinants for time-of-day sensitivity shown in Fig. 5, the metrics of the main oscillatory component were derived using a fixed-ridge threshold of 40. This ridge threshold value was chosen as it offered a well-balanced threshold suitable for comparing signals of varying strengths.

**Multiresolution analysis.** Detrended time-series data were decomposed into a set of different wavelet details $D_j$ representing distinct disjoint frequency bands and a final smooth using a discrete wavelet transform-based multiresolution analysis[52]. The algorithm has been implemented using the 'PyWavelets' python package[53], with a *db20* wavelet of the Daubechies wavelet family as previously described in Myung, Schmal and Hong et al.[54]. Since each wavelet detail $D_j$ represents a period range between $2^j \Delta t$ and $2^{j+1} \Delta t$ (for j = 1, 2, 3, …) we downsampled the time series from a Δt = 10 min to a Δt = 30 min sampling frequency to obtain a circadian period band between 16–32 h for further analysis. Since the MRA decomposes the variance of the detrended signal with respect to the different disjoint period bands, it can be used to determine the rhythmicity of the signal in the circadian period range[55].

**Global circadian strength.** To calculate the global circadian strength (*GCS*) of a cell line model *i*, the autocorrelation peak value

(peak), the wavelet-based continuous ridge length (ridge), and the discrete circadianicity component (circadianicity) were normalized to the respective maximum (max) value measured among all tested cell line models and averaged according to the following equation:

$$GCS_i = \text{mean}\left(\frac{peak_i}{peak_{max}}, \frac{ridge_i}{ridge_{max}}, \frac{circadianicity_i}{circadianicity_{max}}\right) \quad (1)$$

**Statistical analysis.** Linear regression model fitting of *Bmal1* and *Per2* circadianicity components obtained by MRA was done with the 'fitlm' MATLAB function. Significant variances in circadian parameters between wild-type U-2 OS cells and both *Cry1*-sKO and *Cry1/2*-dKO cells were calculated with a one-way ANOVA and Tukey's post-hoc test using the 'anova1' and 'multcompare' MATLAB functions.

## Multi-parametric analysis of growth dynamics

Growth data obtained from long-term live-cell imaging was smoothed using a robust local regression approach of weighted linear least squares and a 2nd degree polynomial model ('rloess' MATLAB function). Doubling times of smoothed cell numbers or confluency were calculated according to the following equation:

$$\text{Doubling time}(t) = t * \frac{\log(2)}{\log(y_0/y_t)} \quad (2)$$

where $t$ refers to the time of assessment, in our case 96 h, and $y_0$ refers to the cell number at timepoint 0. To calculate the exponential growth rate $k$ per unit of time $t$, we normalized cell numbers to the initial timepoint 0 ($y_0$) and fitted an exponential function to the growth curves:

$$\text{Growth}(t) = y_0 * e^{(k*t)} \quad (3)$$

Exponential function fitting was done with the MATLAB 'fit' function[51]. Linear regression model fitting to different combinations of growth parameters was done with the 'fitlm' MATLAB function.

## Estimation of drug sensitivity parameters

Drug response data obtained from long-term live-cell imaging was smoothed using a robust local regression approach of weighted linear least squares and a 2nd degree polynomial model ('rloess' MATLAB function). Following the method established by Hafner et al.[30], we computed the growth rate inhibition (GR) at time $t$ and for each dose $c$ as follows:

$$GR(c,t) = 2^{k(c,t)/k(0)} - 1 \quad (4)$$

where $k(c,t)$ is the growth rate under drug treatment and $k(0)$ is the growth rate of untreated cells. Drug response parameters were retrieved by fitting the dose-dependent GR-values to a sigmoid curve using the following equation:

$$GR(c) = GR_{inf} + \frac{1 - GR_{inf}}{1 + (c/GEC_{50})^{h_{GR}}} \quad (5)$$

where the fitted parameters are as described in Hafner et al.[30]. Standard $EC_{50}$-values were calculated by fitting final nucleus counts, normalized to the respective count of the control, to the following sigmoidal function:

$$f(c) = E_{min} + \frac{1 - E_{min}}{1 + (c/EC_{50})^h} \quad (6)$$

where $E_{min}$ corresponds to the minimum response, restricted to values between 0 and 1, and $h$ is the hill slope of the response curve, constrained to 0.5–10. Sigmoidal function fitting steps were done with the MATLAB functions 'fit' (GR metrics) or 'lsqnonlin' ($EC_{50}$ value)[51]. Hierarchical clustering analysis was implemented with the MATLAB 'clustergram' function using an Euclidian distance and average linkage method[51]. To account for the different tested dose ranges across drugs for clustering, $GEC_{50}$-values were normalized to the dose at $GR_{inf}$. Pearson's linear correlation coefficients across drug sensitivity metrics were computed using the MATLAB 'corr' function. For the comparison of cellular growth dynamics to drug doses evoking the maximum tested effect, we chose growth curves corresponding to the doses closest to the determined $GR_{inf}$-values.

## Time-of-day sensitivity evaluation

Drug response data of time-of-day treatment experiments was smoothed using a moving average ('smoothdata' MATLAB function). Final nucleus counts were normalized to the nucleus count at the respective time of treatment. ToD response data from U-2 OS WT and *Cry1/2*-dKO cell lines were obtained from confluency readouts in the brightfield channel. Time-of-day response curves were generated from the relative final responses of each treatment timepoint to the final response at timepoint 0 and interpolated using the 'smoothing spline' function in MATLAB. The smoothing parameter was set to 0.7. The maximum range across smoothed time-of-day responses ($ToD_{MR}$) was calculated by subtraction of the minimum from the maximum relative response. To assess whether the smoothing of response data introduces artifacts into $ToD_{MR}$ estimates, we performed linear regression analysis between $ToD_{MR}$-values from raw and spline smoothed data using the 'fitlm' MATLAB function and confirmed a high positive correlation (Supplementary Fig. 7a). $ToD_{MR}$-values were clustered as described for the drug sensitivity parameters, with missing data points substituted by values from the closest relevant column, adhering to the 'nearest-neighbor' principle. Polar histograms of the treatment times with maximum and minimum benefit were generated using the 'polarhistogram' MATLAB function with 'probability' normalization to depict each bar's height as the fraction of observations within its bin relative to the total observations.

## Correlation and shapley value regression analysis

Linear regressions of cell-intrinsic features and drug-dependent ToD sensitivity have been obtained using the 'stats' and 'optimize' modules of the 'SciPy' and the 'uncertainties' package of the Python programming language. Predictions of $ToD_{MR}$-values from new data was based on fitting a linear regression model to the original data that was associated with Pearson correlation coefficients ≥0.5, utilizing the 'fitlm' MATLAB function. Bland-Altman plots comparing predicted and actual $ToD_{MR}$-values were generated using the 'Bland-Altman and Correlation Plot' package (v1.12.0.0) from MATLAB Central File Exchange (2017, Ran Klein). Live-imaging data yielding growth, drug sensitivity, and ToD sensitivity metrics from the new cell lines utilized for the predictions were based on the confluency readout due to the unviability of nuclear reporter cell lines for most models. Only the drug sensitivity metrics for MCF7 were based on nuclear counts. The relative importance of the different parameters in a multiple linear regression model is obtained by Shapley value regression via the Python 'dominance-analysis' package.

## Circadian clock gene expression analysis

Gene expression data of circadian clock genes were obtained from the Cancer Cell Line Encyclopedia Dependency Map (CCLE DepMap, https://sites.broadinstitute.org/ccle/datasets, Q4 of 2022)[56]. Circadian clock genes of the individual breast cancer cell lines were clustered using the 'clustermap' plotting function of 'seaborn' with the euclidian distance metric and complete linkage method. Spearman rank

correlation coefficients of circadian clock genes and drug-specific ToD sensitivity values were computed using the Spearman rank correlation algorithm of the 'SciPy stats' Python language module and hierarchically clustered with the correlation distance metric and single linkage method. Regression plots were obtained using the 'jointplot' plotting function from 'seaborn', showing drug-specific correlation between the ToD sensitivities and gene expression of each cell line, including their distribution, squared spearman rank correlation coefficient, p-value, and a confidence interval of 95%.

**Supervised and unsupervised dimensionality reduction**

Supervised dimensionality reduction by linear discriminant analysis was performed with the 'LinearDiscriminantAnalysis' function of the 'sklearn discriminant_analysis' Python language module. The default parameters were retained, using the exact full singular value decomposition solver and 1 component. Median-based binarization of the drug-dependent ToD sensitivity values was used as a target, and the resulting linear discriminant vector was plotted using random y-axis units to avoid overlapping of the data points. The robustness of the discriminative information from the LDA was tested using a leave-one-out cross-validation (LOOCV) strategy (Supplementary Fig. 7b). LOOCV was conducted using the 'model_selection' module from 'sklearn' in Python (v3.9.7) and implemented via the PyCharm Community Edition IDE (v2021.2.2). The contribution to the obtained discriminative information is shown in percentage for each of the circadian clock genes. Unsupervised dimensionality reduction by principal component analysis (PCA) was performed with the 'PCA' function of the 'sklearn decomposition' Python language module using the exact full singular value decomposition solver and number of components equivalent to the sample size of the corresponding drug. Further analysis of the most informative PCA components was done by drawing the biplots of the first two principal components and showing the loadings of each circadian clock gene for both components.

**Reporting summary**

Further information on research design is available in the Nature Portfolio Reporting Summary linked to this article.

## Data availability

The experimental raw data and data tables generated in this study have been deposited in the Figshare database under the identifier https://figshare.com/projects/Time-of-Day-Drug-Response/180916. Source data are provided in this paper.

## Code availability

All code used for the data analysis in this work (in MATLAB and Python) is publicly available through the dataset repository Zenodo under the identifier https://zenodo.org/doi/10.5281/zenodo.11656060.

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

## Acknowledgements

We thank Malthe Skytte Nordentoft Nielsen, as well as the laboratories of Michela Di Virgilio and Ingeborg Tinhofer-Keilholz, for their valuable feedback on our project. We are grateful to Annika Winkler and Marie Möser for their assistance with sample preparation, as well as to Sofía Peso-García and Franziska Reiher for their experimental support. Lastly, we thank Christian Gabriel and Valentina Alejandra Balde Araya for valuable feedback during the review of this manuscript. The results are part of a project funded by the German Federal Ministry of Education and Research (BMBF) through the e:Med Juniorverbund DeepLTNBC TP3-01ZX1917C. C.E. was partially supported by the Deutsche Forschungsgemeinschaft (DFG, German Research Foundation)–RTG2424/ CompCancer – project number: 377984878 and is enrolled in the doctoral program of the Berlin School of Integrative Oncology (BSIO). C.S. acknowledges support from the DFG–SCHM 3362/4–1 project number: 511886499.

## Author contributions

C.E., A.K., and A.E.G. conceived and planned the experiments. C.E. performed the experiments. F.M.M. performed bioluminescence recordings of three cell lines. C.E., C.S., J.D., and S.D.L. analyzed the data. A.M.F. and A.K. supported the planning and implementation of the bioluminescence recordings. C.E., C.S., A.M.F., J.D., S.D.L., J.S., T.S., U.K, H.H., A.K., and A.E.G. contributed to the interpretation of the results. C.E. and A.E.G. wrote the manuscript. All authors provided critical feedback and helped shape the research and manuscript.

## Funding

## Competing interests
