## [Peer Review File · Nature Communications]

Reviewers' Comments:

Reviewer #1:

Remarks to the Author:

In "Time-of-day effects of drugs revealed by high-throughput deep phenotyping", Ector and colleagues present a very interesting and comprehensive study of how the molecular circadian clock controls response to therapeutics in cancer cells. They begin by performing deep circadian phenotyping of a wide panel of cancer cell lines and note that there is a range of circadian function in these lines: some have intact and robust rhythms while other hardly have rhythmicity at all. They then switch to introducing their method of drug screening, which is based on tracking growth and cell number over time rather than endpoint assays. The Authors explain how this methodology may more accurately capture drug sensitivity than endpoint approaches. Using a panel of breast cancer cell lines, the Authors next demonstrate that many lines exhibit a time of day response to cancer, and that this bears some correlation with their circadian metrics (particularly the amplitude of Bmal1 promoter oscillation). Finally, the Authors attempt (with limited success, in my view), to correlate time of day dependent response of drugs to differences of expression in specific molecular clock genes. The Authors conclude that effective usage of chronomedicine requires circadian phenotyping the cells, and that their novel methodology may point the way towards understanding the heterogeneity of circadian function in cancer and turning this into action in the clinic.

Overall, I am very enthusiastic about this approach, and the depth to which the Authors went to gain detailed metrics on rhythms in cancer cells and time of day drug response and correlated these. This approach could be the beginning of a pipeline to phenotype cancers that can make a big impact in the future. Many of my comments below are highly critical, and are designed to address some experimental and analytic shortcomings I identified. Perhaps more importantly, the paper as currently written will not make as big an impact as it should because much of the description of the methods and the rationale behind them is overly complex and inscrutable, and will not be comprehensible to those who are unfamiliar with these fields, nor is the writing appropriate for a broad multidisciplinary journal. My major and minor points are below.

Major points

- This is a manuscript that uses new and exciting methodology to evaluate the time-dependent effects of drugs on cancer cells, and yet the word "cancer" is neither in the title nor the abstract. This means that a large portion of the audience that most needs to read this, cancer biologists, may miss it. The Authors must change the title and abstract to include the fact that this is a cancer study.
- The results sections for Figures 2 and 3 need to be rewritten in much more basic language for non-experts, or I fear that this paper will be incomprehensible to most readers. At risk of revealing myself, I am probably the Authors' ideal reader: a circadian rhythms and cancer researcher who has a colleague in my Department that performs high-throughput drug testing. Despite this prior expertise, I had a hard time understanding the reasoning for many of the different analysis methods in Figures 2 and 3, which means that non-experts might be completely lost. I would recommend that the Authors have a colleague who is not an expert in either circadian rhythms or high throughput screening read these sections so the Authors can better understand which methods require high-level explanation for non-experts.
- If I understand Figure 2 correctly, many of the circadian metrics (autocorrelation, lag 2nd peak, ridge length) average the Per2::Luc and Bmal1::Luc signal, despite the fact that Bmal1 tends to be a lower-amplitude oscillator (as seen in 2c), and that Bmal1 oscillations arise from the ROR/REV-ERB arm while Per2 oscillations arise from the core CLOCK/NPAS2-BMAL1 arm. The Authors should explain why it is biologically meaningful to calculate circadian metrics from the average of both these signals, rather than consider them separately. This is especially important because in Figure 5, the Authors demonstrate that Bmal1 amplitude is the most significant predictor of time of day drug sensitivity, calling into question the accuracy and scientific appropriateness of this approach in Figure 2.

- One potential flaw of generating Bmal1::Luc and Per2::Luc reporter cells by lentivirus is that the lentivirus inserts randomly into the genome (as opposed to, say, the Per2-Luc animal model, which is a fusion protein at the Per2 locus). Because of this, integration site of the reporters into the genome matters: if they 'land' in a heterochromatic site, this may lead to a lower signal or noise. Therefore I am not 100% confident in the Authors only relying on reporter luciferase assays to assess circadian rhythms in their cells. Since the potential molecular clock of almost all of these cells was not previously characterized, I would like to see the Authors 'spot-check' their findings by picking two previously unreported lines, one with robust rhythms (as shown by the promoter luciferase assays) and one with very weak rhythms, and perform a 4-hour resolution, 48-hour time-series assay with 3 biological replicates to assess molecular clock transcript oscillation by qPCR. If this recapitulates their findings with the promoter luciferase, it will lend confidence to the rest of the results.
- For most of the Manuscript, the Authors use a panel of breast cancer and normal breast epithelium cells. They apparently generated circadian reporter lines for these cells, but never included the complete circadian metrics of these lines in the paper, as they did for their pan-cancer panel in Figure 2. This is extremely important: if a breast cancer line has a very weak or absent circadianity, it would make the time of day response to drugs results in that line very difficult to interpret or even biologically meaningless (or connected to cell cycle instead of circadian rhythms). Since we need to know this information to fully evaluate Figures 4 and 5, please report on the complete circadian metrics for the entire breast cancer panel.
- For Figure 6, I disagree with the interpretation of the Authors on these data. They showed nicely in Figure 5 that clock metrics calculated in Figure 2 (and remaining to be shown) correlated with time of day dependent response. In Figure 6, the way I see the data, there is a remarkable lack of correlation between time of day dependent response and expression of specific clock genes, especially the main transcription factors BMAL1, CLOCK, and NPAS2. Overall, there is not a clear trend correlating molecular clock gene expression overall with TODmr. It is not clear that expression differences in any single molecular clock gene have anything to do with circadian metrics they calculated since almost every gene except for BMAL1 has at least 1 paralogue. I also do not understand Figures 6D, E, and F: if there is not a statistically significant correlation between a given clock gene in a given cell line and TODmr, it does not make sense to rank and score these non-significant correlations in 6D. Similarly, in E and F, as far as I can tell from the Methods, LDA analysis does not include statistical testing, so these data are not necessarily meaningful, especially since the overall contribution of each clock gene to response is ~7% or less. I would recommend that the Authors rewrite the text describing Figure 6 to better reflect the results. A lack of correlation between specific clock genes and TODmr in no way detracts from the strength of the other findings.

Minor points

- Citation 3, Mure et al Science 2018 reports that 80% of all protein coding genes are rhythmically expressed, not 40% that is written in this manuscript. If the Authors disagree with the analysis methodology and did their own analysis to arrive at the 40% number, they must state that in the manuscript. Otherwise, the 40% number is a misrepresentation of the Mure et al paper and must be corrected.
- The signal decomposition approach in Figure 2i and 2j, and the high resolution data the Authors generated with the reporter systems, gives the Authors a chance to understand how rhythms other than circadian rhythms may influence drug response. However, in Figure 2j, the Author's approach of comparing circadianity to noise without considering other components seems like a missed opportunity, since the influence of 12-hour rhythms can just as easily influence drug response as 24-hour rhythms. For instance, a recent 2017 Cell Metabolism publication (PubmedID 28591634) discussed a mechanistic basis for 12 hour rhythms in mammalian cells. This might explain the results in HCC1937 in Figure 4i. I do not envision that the Authors need to do additional experiments for this, but it should be addressed in the Discussion.
- On page 6, line 178, the Authors state "To showcase our approach's ability in detecting within-tissue differences...". The Authors are incredibly far away from adapting this approach from using established and immortalized cell lines that grow in two-dimensions to detecting within-tissue

differences. Please soften or change this language.

- In Figure 3, the Authors use both brightfield and nuclear staining to assess cell number. Why is it appropriate to use both, rather than nuclear staining only? Nuclear staining is the standard in the field due to its accuracy, and the enormous differences between the brightfield and nuclear signal in Supplemental Figure 2 suggest that the brightfield is not very accurate in some of the cell models.
- It was not initially clear why the Authors favored normalized growth rate inhibition in Figure 3 to assess drug sensitivity until I read the Hafner et al paper. I would suggest the Authors re-read that paper, especially the Abstract and Introduction, and adapt some of the language of that paper to better explain the rationale behind their approach, as compared to endpoint testing that is more common in the field.
- Considering the rich literature on time-of-day dependent response to 5-FU and platinum-based chemotherapies, it would be more appropriate for the Authors to cite some key papers on these drugs rather than just a single review.
- On Page line 308-309, the Authors state that "While the average ToD variability highlights the benefits of a single model (Fig. 4g), the chronotherapeutic index reveals distinctions between cancer and non-malignant models". This is an over-interpretation of their data, considering the Authors are only using a single non-malignant line that is still immortalized and adapted to 2D culture. If the Authors want to make a blanket statement like this, they must bring in additional non-transformed models that more closely recapitulate human biology, such as Human Mammary Epithelial Cells (HMECS, ATCC PCS-600-010), otherwise they should alter sweeping statements like this based on comparisons to only MCF10a.

Reviewer #2:

Remarks to the Author:

Review of NCOMMS-23-53863-T

This manuscript describes a set of experimental and computational tools to create phenotypic profiles of cell lines in culture with respect to: a) the strength and the properties of circadian regulation; b) cell proliferation dynamics (both unperturbed and in the presence of a set of anti-cancer drugs); and c) the presence and the strength of Time of Day (ToD) effects of anti-cancer drugs.

The authors used this framework to describe the properties of circadian rhythms in 9 cancer cell lines of different origin using Bmal1-Luc and Per1-Luc reporters. The results of these experiments indicate that different cell lines have substantial variability in the strength of circadian clock regulation. In addition, they tested cell proliferation dynamics and ToD effect for 9 breast epithelial cell lines (1 non transformed and 8 transformed cancer cell lines) treated or untreated with 7 anti-cancer drugs. The results of these experiments indicate that there is a substantial amount of variability in proliferation dynamics for different cell lines, and that they are differentially sensitive to the different anti-cancer drugs. As for ToD effects, the authors show that different compounds have different ToD in different cell lines.

The manuscript then describes several statistical models that try to explain ToD effects for different compounds using the circadian clock and drug sensitivity metrics described in the previous experiments. The authors observe high correlation between the amplitude of the oscillations and other parameter of the circadian clock reporters for some of the compounds. Finally, the conduct a reanalysis of RNA-seq gene expression data for 13 circadian clock regulators in the cell lines used in the previous experiments. The results of the reanalysis indicate that ToD effects are not correlated to the expression levels of a single circadian clock regulator. Rather, gene expression differences of several different regulators contribute to differences in ToD effects between different cell lines, and between different compounds.

The manuscript is well written, the experimental methods employed are explained to a good level of detail, and the results of the experiments are well presented. In addition, I believe that dissecting the molecular pathways of the circadian clock in different cellular backgrounds, and identifying the factors behind differential drug sensitivities and ToD effects in different cancer cell lines is of high interest to a large audience. Having a diverse set of measurement tools and an analytical framework to integrate these measurements, as described in the manuscript, could produce mechanistic hypothesis to be experimentally tested. Alternatively, these measurements and models could be used to generate predictions of drug sensitivities and ToD on yet untested cell lines.

In its current form, though, the findings in the manuscript are almost exclusively descriptive. Overall, they do not deliver on the potentially interesting applications of an experimental/analytical platform to understand and forecast ToD effects in different cancer cell line backgrounds. I recommend major revisions and resubmission to meet this potential. I am detailing my major and minor concerns below.

Major concerns:

1) The results presented mostly just correlate circadian clock properties and proliferation dynamics with drug sensitivities and ToD effects. Similarly, the gene expression analysis only broadly establishes correlations between gene expression of circadian clock regulators and ToD effects, but it does not shed light on possible mechanisms behind the observed correlations. I understand that it would be unfeasible to experimentally test all the positive/negative correlations between circadian clock parameters and drug sensitivities/ToD effects. On the other hand, the authors should present the results of at least one experiment where the cell lines are experimentally perturbed. As an example, what would happen to ToD effects of one of the drugs in a certain cell line when Bmal1 and/or Per1 are knocked out by CRISPR?

2) The experimental/analytical platform described in the manuscript is used to measure statistical associations between predictor variables (Circadian clock features, proliferation dynamics, etc.) and response variables (Drug sensitivities, ToD effects). The predictive value of these associations is not tested in cell lines that were not used to derive the statistical associations. To show real predictive value of these models, the authors should measure the difference between Drug Sensitivities and ToD values predicted by their model and actual measured values in cell lines that were not used to calculate the statistical associations. If the models described in the manuscript have high predictive value, the difference between predicted and measured values for the response variables should be small.

3) Throughout the manuscript, all the replicates are described as "technical", and there are no mentions of biological replicates. While this might just be a matter of defining "technical" vs. "biological" replicates when using cell lines, can the authors please specify if the replicates were run on different days? In my opinion, independent end-to-end replicate experiments run on different days with the same cell line are sufficient to qualify as biological in this context.

Minor concerns:

4) Fig. 1 does not contain information about experimental results and is highly redundant with other panels of other figures. As such, it seems more fit for a graphical abstract than as a manuscript figure. In addition, I seldom observe figures being mentioned in the introduction section of a manuscript. I would remove Fig. 1 from the manuscript.

5) All throughout the manuscript, when p-values are reported, please specify which comparisons/tests were taken in consideration for each statistical test. Also, I suggest the authors consult a statistician on whether running multiple pairwise t-tests when having multiple comparisons, rather than an ANOVA test with a post-hoc test, is the appropriate approach for testing statistical significance with more than two experimental conditions.

6) The Github repo link provided in the manuscript gives me a 404 error when trying to navigate that webpage. Please verify that the link is functional. It would really benefit the community if the code base used to generate the results presented in this manuscript could be reutilized for future projects.

7) While the authors provide a link to the code base for the analysis (See point 5 above), in the data availability section they mention that "The experimental time series data and data tables for all results of this study are available upon request" in the Data Availability section. The manuscript is highly computational in nature, and a lot of the results presented here depend on fairly sophisticated analytical pipelines. In the interest of promoting FAIR principles (Which are often required by funders and journals nowadays), I recommend the authors deposit the primary data underlying their manuscript in a generalist repository, such as Zotero, Figshare, or Biostudies.

8) Fig. 2D: How was the 95.4% CI calculated? Optional: While all confidence intervals are arbitrary, is there a particular reason why 95.4% was chosen as a threshold, as opposed to the more common 95% value for CIs?

9) Page 5, Line 130: "Heterogeneity between and within cancer entities was further observed for the oscillation period [...]". What do the authors mean with "cancer entities"? Is it cancer type of origin? Or is it cancer cell line? Please specify.

10) Page 6, Line 177; and Fig. 3D "To showcase our approach's ability in detecting within-tissue differences, we examined nine cell lines of the triple-negative breast cancer (TNBC) subtype alongside the non-malignant MCF10A breast cell model." What do the different groups (BL1, BL2, MES) represent? Different stages of breast cancer progression of the tissue from which the cell lines were generated? Please specify and briefly describe in the main text and not only in the abbreviations.

11) Fig. 3E: What does the shaded area represent? Please specify in the legend. Assuming it is a representation of the error or a CI of the mean of multiple measurements (i.e. the dots in the plot represent a mean of multiple measurements), please also specify the number of replicates used to calculate the mean and the interval.

12) Fig. 3, Legend: "Data in c-f represents the mean \pm s.e.m. from 9 snapshots. k-value errors reflect the 95% CI." What does the term "snapshot" represent here? Are these 9 independent experiments run on different days? Are 9 cell dishes run in parallel on the same day? If either of these guesses are correct, please use "replicates" and define if these are technical or biological (cfr. point 3).

13) Fig. 4A: I believe that in the right panel the color scheme of the arrows after "sorting" is incorrect, based on the length of the treatment. The red arrow after "sorting" should be the shortest (24 hrs, or 32 - 8 hrs, based on the experimental time above), then blue (28 hrs, or 32 - 4 hrs, based on the experimental time above), then green (32 hrs, or 32 - 0 hrs, based on the experimental time above), then the others, which are already in the correct order. Please double check.

14) Page 13, Line 393: "Importantly, the contribution of each gene in discriminating between ToD MR responses varied for the different drugs, suggesting that each drug interacts differently with the molecular components of the circadian clock (Supplementary Fig. 6b-h)." An alternative explanation for this observation is that expression levels of the target(s) of the drug exhibit circadian variation. There is no evidence in the manuscript that the anti-cancer drugs used here (Which have substantially different cellular targets) interact directly with circadian clock regulators. To rule out that circadian oscillation of expression levels of the targets underlies the variability of ToD among different drugs, the authors should perform qRT-PCR quantification of mRNA expression and/or western blots for a panel of these targets at different clock times.

15) Page 14, Line 437" "Notably, in agreement with recent studies³⁹ , our results indicate that the sensitivity metrics exhibit milder correlations, challenging the conventional binary classification of sensitive versus resistant models and suggesting that drug sensitivity rankings are rather metric-specific and drug dependent (Fig. 3)." Please specify which "sensitivity metrics" you are referring to. Also, "milder correlations" compared to which treatment/conditions/etc?

Reviewer #3:

None

We would like to express our appreciation to the reviewers for dedicating their time to review our manuscript and for providing us with valuable comments and feedback.

In this response, we thoroughly address each of the raised concerns, incorporating new experiments and analysis to our new version of the manuscript. To facilitate tracking the changes, we are providing a highlighted version of our revised manuscript. Changes made independently of the reviewer's feedback are described at the end of this letter.

****REVIEWER #1** – Circadian rhythms, cancer (Remarks to the Author):**

In “Time-of-day effects of drugs revealed by high-throughput deep phenotyping”, Ector and colleagues present a very interesting and comprehensive study of how the molecular circadian clock controls response to therapeutics in cancer cells. They begin by performing deep circadian phenotyping of a wide panel of cancer cell lines and note that there is a range of circadian function in these lines: some have intact and robust rhythms while other hardly have rhythmicity at all. They then switch to introducing their method of drug screening, which is based on tracking growth and cell number over time rather than endpoint assays. The Authors explain how this methodology may more accurately capture drug sensitivity than endpoint approaches. Using a panel of breast cancer cell lines, the Authors next demonstrate that many lines exhibit a time of day response to cancer, and that this bears some correlation with their circadian metrics (particularly the amplitude of Bmal1 promoter oscillation). Finally, the Authors attempt (with limited success, in my view), to correlate time of day dependent response of drugs to differences of expression in specific molecular clock genes. The Authors conclude that effective usage of chronomedicine requires circadian phenotyping the cells, and that their novel methodology may point the way towards understanding the heterogeneity of circadian function in cancer and turning this into action in the clinic.

Overall, I am very enthusiastic about this approach, and the depth to which the Authors went to gain detailed metrics on rhythms in cancer cells and time of day drug response and correlated these. This approach could be the beginning of a pipeline to phenotype cancers that can make a big impact in the future. Many of my comments below are highly critical, and are designed to address some experimental and analytic shortcomings I identified. Perhaps more importantly, the paper as currently written will not make as big an impact as it should because much of the description of the methods and the rationale behind them is overly complex and inscrutable, and will not be comprehensible to those who are unfamiliar with these fields, nor is the writing appropriate for a broad multidisciplinary journal. My major and minor points are below.

We are grateful for the reviewer's enthusiasm about our approach. Following the reviewer's suggestion, we have now added explanatory sentences and rewritten several sections of our manuscript, to better describe the methods and the rationale behind them with the hope to improve the readability of our manuscript. Below, we address each major and minor point of the reviewer individually.

Major points:

1. This is a manuscript that uses new and exciting methodology to evaluate the time-dependent effects of drugs on cancer cells, and yet the word “cancer” is neither in the title nor the abstract. This means that a large portion of the audience that most needs to read this, cancer biologists, may miss it. The Authors must change the title and abstract to include the fact that this is a cancer study.

This is a great suggestion. We now included the word cancer in the title and abstract (lines 53 and 59) accordingly.

2. The results sections for Figures 2 and 3 need to be rewritten in much more basic language for non-experts, or I fear that this paper will be incomprehensible to most readers. At risk of revealing myself, I am probably the Authors' ideal reader: a circadian rhythms and cancer researcher who

has a colleague in my Department that performs high-throughput drug testing. Despite this prior expertise, I had a hard time understanding the reasoning for many of the different analysis methods in Figures 2 and 3, which means that non-experts might be completely lost. I would recommend that the Authors have a colleague who is not an expert in either circadian rhythms or high throughput screening read these sections so the Authors can better understand which methods require high-level explanation for non-experts.

We thank the reviewer for bringing this to our attention. Following the reviewer's suggestion we have shared this manuscript with a set of colleagues of mixed scientific background and in the new version of the manuscript we incorporated their suggestions to improve the understanding of our manuscript. We now describe results and the rationale behind the approaches to Figure 2 and 3 using a more explicit, general, and basic language aimed to address a broader readership. See highlighted text in the Results section on pages 4–8.

3. If I understand Figure 2 correctly, many of the circadian metrics (autocorrelation, lag 2nd peak, ridge length) average the *Per2::Luc* and *Bmal1::Luc* signal, despite the fact that *Bmal1* tends to be a lower-amplitude oscillator (as seen in 2c), and that *Bmal1* oscillations arise from the ROR/REV-ERB arm while *Per2* oscillations arise from the core CLOCK/NPAS2-BMAL1 arm. The Authors should explain why it is biologically meaningful to calculate circadian metrics from the average of both these signals, rather than consider them separately (3a). This is especially important because in Figure 5, the Authors demonstrate that *Bmal1* amplitude is the most significant predictor of time of day drug sensitivity, calling into question the accuracy and scientific appropriateness of this approach in Figure 2 (3b).
 - a. We appreciate the concerns raised regarding the averaging of *Per2::Luc* and *Bmal1::Luc* signals, given their distinct oscillatory behaviors and biological pathways. As the reviewer suggests, there is no a-priori reason to claim that averages of *Per2* and *Bmal1* are more biologically meaningful than individual metrics. In Figure 2 we include individual results for *Per2* and *Bmal1* signals, color-coded in blue and yellow, respectively. Only for simplicity of presentation, we averaged these values to generate the rankings of the means as shown in Fig. 2b,e,h and k. Addressing the reviewer's concern, we have now independently analyzed and ranked *Bmal1* and *Per2* signals and included the results in Supplementary Fig. 1e and f.
 - b. Furthermore, we have made two additions to the text, highlighting the significant distinctions that may arise from separately analyzing *Bmal1::Luc* and *Per2::Luc* signals. These modifications are present in the Results section, on page 6, lines 169-174, as well as in the Discussion section, on page 17, lines 511-515. Additionally, we have expanded on the importance of conducting separate analyses for these signals, particularly emphasizing *Bmal1*'s unique role in predicting time-of-day drug sensitivity, as suggested by the reviewer, and as illustrated in Figure 5.
4. One potential flaw of generating *Bmal1::Luc* and *Per2::Luc* reporter cells by lentivirus is that the lentivirus inserts randomly into the genome (as opposed to, say, the *Per2*-Luc animal model, which is a fusion protein at the *Per2* locus). Because of this, integration site of the reporters into the genome matters: if they 'land' in a heterochromatic site, this may lead to a lower signal or noise. Therefore I am not 100% confident in the Authors only relying on reporter luciferase assays to assess circadian rhythms in their cells. Since the potential molecular clock of almost all of these cells was not previously characterized, I would like to see the Authors 'spot-check' their findings by picking two previously unreported lines, one with robust rhythms (as shown by the promoter luciferase assays) and one with very weak rhythms, and perform a 4-hour resolution, 48-hour time-series assay with 3 biological replicates to assess molecular clock transcript oscillation by qPCR. If this recapitulates their findings with the promoter luciferase, it will lend confidence to the rest of the results.

Thank you for raising this very important point regarding the potential variability of reporter luciferase assays in assessing cellular circadian rhythms due to the random integration of lentivirus-based *Bmal1*- and *Per2*-Luc reporters. Indeed, a limitation of our current approach is that it does not neglect the possibility of random insertions and does not distinguish them from those that truly report the circadian nature of the cell line. We have now included in the Discussion section a statement describing the limitation of our approach (page 18 lines 535–538).

The generation of the circadian reporter cell lines was done in collaboration and implementing the protocols from the Kramer lab (Laboratory of Chronobiology, Charité) who has extensive expertise in the generation of circadian reporter cells lines. We gain confidence in our current approach from the following experiences:

- i. Some of the reporter cell lines we developed, namely MCF10A, MCF7, and U-2 OS WT and KO cell lines, were independently generated in other laboratories. This enabled us to cross-reference our evaluations of the circadian clock, revealing a strong alignment with previous research findings¹⁻⁴.
- ii. At the initial phase of this project, we were especially concerned about this problem and repeated the *Bmal1*-Luc reporter insertions into a set of wild-type cells. The repetition aimed to assess the consistency of our protocol showed consistent luciferase signals.
- iii. All our recordings are from non-clonal cell lines and at the population level, where we expect the effects of potential lentiviral misintegrations to be reduced through population averaging effects.

However, despite this experience, we acknowledge that the possibility of adventitious insertions cannot be completely ruled out. We agree with the reviewer that performing additional confirmatory qPCR analyses would increase confidence in our results. Such an approach would undoubtedly be valuable for subsequent research efforts seeking a more thorough and mechanistic understanding of the time-of-day effects on the previously unreported cell lines. Nonetheless, it is beyond the scope of our current study.

5. For most of the Manuscript, the Authors use a panel of breast cancer and normal breast epithelium cells. They apparently generated circadian reporter lines for these cells, but never included the complete circadian metrics of these lines in the paper, as they did for their pan-cancer panel in Figure 2. This is extremely important: if a breast cancer line has a very weak or absent circadianity, it would make the time of day response to drugs results in that line very difficult to interpret or even biologically meaningless (or connected to cell cycle instead of circadian rhythms). Since we need to know this information to fully evaluate Figures 4 and 5, please report on the complete circadian metrics for the entire breast cancer panel.

Thank you for bringing this to our attention, as we regret to not having included this information in the previously submitted version. We now included a detailed table, Supplementary Table 2, that outlines all the circadian metrics for the entire breast cancer panel featured in our manuscript. We further refer to that table in the Results section to Figure 5 on page 12, lines 366-367 of the revised manuscript.

6. For Figure 6, I disagree with the interpretation of the Authors on these data. They showed nicely in Figure 5 that clock metrics calculated in Figure 2 (and remaining to be shown) correlated with time of day dependent response. In Figure 6, the way I see the data, there is a remarkable lack of correlation between time of day dependent response and expression of specific clock genes, especially the main transcription factors BMAL1, CLOCK, and NPAS2. Overall, there is not a clear trend correlating molecular clock gene expression overall with TODmr (6a). It is not clear that expression differences in any single molecular clock gene have anything to do with circadian metrics they calculated since almost every gene except for BMAL1 has at least 1 paralogue (6b). I also do not understand Figures 6D, E, and F: if there is not a statistically significant correlation between a given clock gene in a given cell line and TODmr, it does not make sense to rank and score these non-significant correlations in 6D (6c). Similarly, in E and F, as far as I can tell from the

Methods, LDA analysis does not include statistical testing, so these data are not necessarily meaningful, especially since the overall contribution of each clock gene to response is ~7% or less (6d). I would recommend that the Authors rewrite the text describing Figure 6 to better reflect the results. A lack of correlation between specific clock genes and TODmr in no way detracts from the strength of the other findings (6e).

We thank the reviewer for their critical review of Figure 6 and its interpretation, which highlighted a need for an accurate description of our results and consequent discussion of the data. In the following, we will address each of the reviewer's concerns to Figure 6 individually.

- a. We agree with the reviewer that our initial interpretation was overly optimistic regarding correlations between circadian gene expression levels and ToD_{MR} values. We have now revised text passages throughout the Results section of Figure 6. Particularly, we state the exact number of significant correlations instead of referring to them as "several" in the Results section on page 14, lines 420-421, and we explicitly mention that the overall correlation was relatively weak on page 14, lines 423-426 and on page 15, line 456 of the revised manuscript.
- b. We thank the reviewer for highlighting the important issue of gene paralogues in our circadian clock gene panel. This part of our study was aimed to explore whether snapshot gene expression data could potentially reflect ToD_{MR} values. We acknowledge that the presence of paralogues for most clock genes indeed complicates the interpretation of how expression of individual genes may reflect ToD_{MR} values. Consequently, we have now elaborated on this point in the revised manuscript, explaining how this genetic redundancy might impact our findings (page 17, line 516-519).
- c. We recognize that presenting non-significant correlations in Figure 6D contribute to confusion and kindly thank the reviewer for pointing this out. Acknowledging the lack of significant correlations, we adapted our statements as discussed above (review point 6a) and moved subplot 6D from the original manuscript to Supplementary Fig. 6b in the revised version of the manuscript.
- d. Indeed, the LDA as we applied it did not initially incorporate statistical testing, which is an important point for consideration. To address this and to enhance the interpretability of our findings, we now conducted a new leave-one-out cross-validation (LOOCV) test on our LDA results. This test provides a quantification of the accuracy in the classification of ToD_{MR} values for four drugs, with accuracies reaching up to 0.89 for torin2 and 0.67 for paclitaxel. These high accuracies suggest a considerable predictive capacity of the LDA model for these drugs. However, the limited sample size of 8 to 9 cell models may have impacted the accuracy for other drugs, indicating that while our approach shows promise for certain analyses, it should be applied with caution, especially when the contribution of individual genes appears small. We have incorporated this additional statistical testing of the LDA in Supplementary Figure 7b of the revised manuscript.

Figure 1 | New leave-one-out cross-validation test (LOOCV) on LDA results discussed in Figure 6 and Supplementary Fig. 6 of the originally submitted manuscript. The color coding reflects the accuracy of each cell line model (*columns*) in being correctly assigned to its ToD-sensitivity group per drug (*rows*), where green and red indicate true and false categorization, respectively. The rightmost column denotes the summed LOOCV accuracy per drug, where 1 refers to 100% accuracy.

- e. We thank the reviewer again for the critical assessment of our results presented in Figure 6. We revised both the Results and Discussion sections related to Figure 6 as described above, aiming to provide a clearer interpretation of our data.

Minor points

1. Citation 3, Mure et al Science 2018 reports that 80% of all protein coding genes are rhythmically expressed, not 40% that is written in this manuscript. If the Authors disagree with the analysis methodology and did their own analysis to arrive at the 40% number, they must state that in the manuscript. Otherwise, the 40% number is a misrepresentation of the Mure et al paper and must be corrected.

We appreciate the reviewer's attention concerning the reported percentages of rhythmically expressed protein-coding genes. We have now corrected the introduction section to accurately reflect that 80% of protein-coding genes show rhythmic expression in primate studies, as reported by Mure *et al.*, *Science*, 2018 (see page 3, line 67 of the revised manuscript).

2. The signal decomposition approach in Figure 2i and 2j, and the high resolution data the Authors generated with the reporter systems, gives the Authors a chance to understand how rhythms other than circadian rhythms may influence drug response. However, in Figure 2j, the Author's approach of comparing circadianicity to noise without considering other components seems like a missed opportunity, since the influence of 12-hour rhythms can just as easily influence drug response as 24-hour rhythms. For instance, a recent 2017 Cell Metabolism publication (PubmedID 28591634) discussed a mechanistic basis for 12 hour rhythms in mammalian cells. This might explain the results in HCC1937 in Figure 4i. I do not envision that the Authors need to do additional experiments for this, but it should be addressed in the Discussion.

We thank the reviewer for the suggestion and the reference to that very exciting publication (PMID 28591634). As proposed by the reviewer we now include this in the Discussion section as an interesting direction for subsequent research (pages 17 and 18, lines 529-534 of the revised manuscript).

3. On page 6, line 178, the Authors state "To showcase our approach's ability in detecting within-tissue differences...". The Authors are incredibly far away from adapting this approach from using established and immortalized cell lines that grow in two-dimensions to detecting within-tissue differences. Please soften or change this language.

We fully agree with the reviewer's comment that our 2D cell line models are not capable to detect within-tissue differences and we apologize for our inaccurate choice of words. To address this and to reflect the limitations of our approach more accurately, we have revised our statement on page 7, now line 190 in the revised manuscript. We further replaced the term "tissue models" with "cell models" throughout the revised version of the manuscript.

4. In Figure 3, the Authors use both brightfield and nuclear staining to assess cell number. Why is it appropriate to use both, rather than nuclear staining only? Nuclear staining is the standard in the field due to its accuracy, and the enormous differences between the brightfield and nuclear signal in Supplemental Figure 2 suggest that the brightfield is not very accurate in some of the cell models.

Thank you for your insightful comment regarding Figure 3. We acknowledge the validity of nuclear staining as the standard method for assessing cell viability due to its accuracy. However, we chose to include confluency results alongside nuclear staining to provide a complementary set of readouts for cell growth, which may be more accessible to some readers. Additionally, we have demonstrated in Supplementary Figure 2b that the doubling times estimated from nuclear counting and confluency exhibit good correlation. We have now explicitly highlighted the advantages and higher accuracy of growth metrics based on nuclear staining in the Results section (page 7, lines 198–201).

5. It was not initially clear why the Authors favored normalized growth rate inhibition in Figure 3 to assess drug sensitivity until I read the Hafner et al paper. I would suggest the Authors re-read that paper, especially the Abstract and Introduction, and adapt some of the language of that paper to better explain the rationale behind their approach, as compared to endpoint testing that is more common in the field.

Following this suggestion, we now adapted the language of the Results section and included a more explicit explanation of the rationale behind our utilization of growth rate inhibition metrics (page 8, lines 228-231 and 239-241).

6. Considering the rich literature on time-of-day dependent response to 5-FU and platinum-based chemotherapies, it would be more appropriate for the Authors to cite some key papers on these drugs rather than just a single review.

We regret not having acknowledged the important key publications in the previous version of our manuscript. We have now included citations to PMID: 2909754, where Sothorn *et al.*⁵ showed circadian stage dependent responses of cancer-bearing rats to doxorubicin and cisplatin. Further, we included citations to PMID: 16928823 and PMID: 7932825, where Wood *et al.*⁶ and Lévi *et al.*⁷ described time-of-day dependent activities of 5-FU (page 16, lines 497-499).

7. On Page line 308-309, the Authors state that "While the average ToD variability highlights the benefits of a single model (Fig. 4g), the chronotherapeutic index reveals distinctions between cancer and non-malignant models". This is an over-interpretation of their data, considering the Authors are only using a single non-malignant line that is still immortalized and adapted to 2D culture. If the Authors want to make a blanket statement like this, they must bring in additional non-transformed models that more closely recapitulate human biology, such as Human Mammary Epithelial Cells (HMECS, ATCC PCS-600-010), otherwise they should alter sweeping statements like this based on comparisons to only MCF10a.

We thank the reviewer for bringing this to our attention. Using a single immortalized non-malignant cell line adapted to 2D culture is indeed a strong limitation of our study and the MCF10A cell line alone does not comprehensively reflect non-malignant models. To better reflect the scope of our study, we adapted the respective text passage on page 12, line 344 of our revised manuscript, where we now particularly refer to the "non-malignant MCF10A cell model" instead of "non-malignant models".

****REVIEWER #2**** — high-throughput deep phenotyping, live-cell (Remarks to the Author):

This manuscript describes a set of experimental and computational tools to create phenotypic profiles of cell lines in culture with respect to: a) the strength and the properties of circadian regulation; b) cell proliferation dynamics (both unperturbed and in the presence of a set of anti-cancer drugs); and c) the presence and the strength of Time of Day (ToD) effects of anti-cancer drugs.

The authors used this framework to describe the properties of circadian rhythms in 9 cancer cell lines of different origin using Bmal1-Luc and Per1-Luc reporters. The results of these experiments indicate that different cell lines have substantial variability in the strength of circadian clock regulation. In addition, they tested cell proliferation dynamics and ToD effect for 9 breast epithelial cell lines (1 non transformed and 8 transformed cancer cell lines) treated or untreated with 7 anti-cancer drugs. The results of these experiments indicate that there is a substantial amount of variability in proliferation dynamics for different cell lines, and that they are differentially sensitive to the different anti-cancer drugs. As for ToD effects, the authors show that different compounds have different ToD in different cell lines.

The manuscript then describes several statistical models that try to explain ToD effects for different compounds using the circadian clock and drug sensitivity metrics described in the previous experiments. The authors observe high correlation between the amplitude of the oscillations and other parameter of the circadian clock reporters for some of the compounds. Finally, the conduct a reanalysis of RNA-seq gene expression data for 13 circadian clock regulators in the cell lines used in the previous experiments. The results of the reanalysis indicate that ToD effects are not correlated to the expression levels of a single circadian clock regulator. Rather, gene expression differences of several different regulators contribute to differences in ToD effects between different cell lines, and between different compounds.

The manuscript is well written, the experimental methods employed are explained to a good level of detail, and the results of the experiments are well presented. In addition, I believe that dissecting the molecular pathways of the circadian clock in different cellular backgrounds, and identifying the factors behind differential drug sensitivities and ToD effects in different cancer cell lines is of high interest to a large audience. Having a diverse set of measurement tools and an analytical framework to integrate these measurements, as described in the manuscript, could produce mechanistic hypothesis to be experimentally tested. Alternatively, these measurements and models could be used to generate predictions of drug sensitivities and ToD on yet untested cell lines.

In its current form, though, the finding in the manuscript are almost exclusively descriptive. Overall, they not deliver on the potentially interesting applications of an experimental/analytical platform to understand and forecast ToD effects in different cancer cell lines backgrounds. I recommend major revisions and resubmission to meet this potential. I am detailing my major and minor concerns below.

We greatly appreciate the time and effort the reviewer has taken to provide a comprehensive review of our manuscript. We are pleased to hear that our writing, experimental methodology, and presentation of results are well-received. In the following, we address each of the reviewer's concerns individually.

Major concerns:

1. The results presented mostly just correlate circadian clock properties and proliferation dynamics with drug sensitivities and ToD effects. Similarly, the gene expression analysis only broadly establishes correlations between gene expression of circadian clock regulators and ToD effects, but it does not shed light on possible mechanisms behind the observed correlations. I understand that it would be unfeasible to experimentally test all the positive/negative correlations between circadian clock parameters and drug sensitivities/ToD effects. On the other hand, the authors

should present the results of at least one experiment where the cell lines are experimentally perturbed. As an example, what would happen to ToD effects of one of the drugs in a certain cell line when Bmal1 and/or Per1 are knocked out by CRISPR?

We agree with the reviewer that while our approach broadly investigates correlations between gene expression of circadian clock regulators and ToD effects, it indeed falls short on exploring possible mechanisms behind observed correlations.

In response to the reviewer's suggestion, we conducted a new series of time-of-day experiments using previously untested cell lines and a panel of three drugs. Specifically, we selected U-2 OS wild-type (WT) and U-2 OS *Cry1/Cry2*-double knockout (dKO) cell lines for these experiments due to their well-established knockout models and consistent circadian characterization, as validated by our deep circadian phenotyping analysis. For these time-of-day drug response experiments, we opted for cisplatin, 5-FU, and paclitaxel, targeting diverse molecular pathways, based on their demonstrated high time-of-day-dependent responses across other cell lines in our previous manuscript version. This approach enabled us to directly evaluate the influence of circadian gene perturbations on time-of-day sensitivities across different drugs.

For all three tested drugs, we observed high ToD-dependent response variations for the WT model, whereas the *Cry1/Cry2*-dKO exhibited substantial reductions in ToD-dependent sensitivity (see Figure 2a). Specifically, the circadian clock perturbed U-2 OS *Cry1/Cry2*-dKO cell model exhibited reduction in ToD-dependent maximum response range (ToD_{MR}) to cisplatin by 62%, by 58% to paclitaxel, and by 40% to 5-FU (see Fig. 2b). Consequently, our new data suggests that an experimental perturbation of the circadian clock negatively affects ToD sensitivity compared to a cell line with a functional circadian clock.

We have incorporated these fresh findings in Supplementary Figure 4c and 4d, accompanied by a thorough explanation on pages 10 and 11, lines 306-315 of the revised manuscript. To facilitate the review of the rebuttal letter, we present them here as Figure 2. We express gratitude to the reviewer for their insightful suggestion, which prompted us to conduct this new set of experiments. We believe these additions substantially enhance the clarity and impact of our manuscript.

Figure 2 | **a**, Overlay of ToD response curves for U-2 OS wildtype and *Cry1/Cry2*-double knock-out cell lines, grouped by the administered drugs. Cell line models are color-coded. Data represents the mean (\pm s.d.) of two plates. **b**, Ranking of the ToD-dependent maximum response range (ToD_{MR}) calculated from the response curves shown in **a**, individually for each drug-cell combination.

- The experimental/analytics platform described in the manuscript is used to measure statistical associations between predictors variables (Circadian clock features, proliferation dynamics, etc.) and response variables (Drug sensitivities, ToD effects). The predictive value of these associations is not tested in cell lines that were not used to derive the statistical associations. To show real predictive value of these models, the authors should measure the difference between Drug Sensitivities and ToD values predicted by their model and actual measured values in cell lines that were not used to calculate the statistical associations. If the models described in the manuscript have high predictive value, the difference between predicted and measured values for the response variables should be small.

We thank the reviewer for the insightful feedback on our analysis on statistical associations between the circadian, growth and drug sensitivity metrics (“cellular factors”) and ToD Maximum Range (ToD_{MR}) values. We have now implemented the following changes:

In response to the reviewer's suggestion, we conducted an extensive examination of various cellular factors' potential to forecast ToD_{MR} values using freshly obtained data. To assess the predictive capability, we initially deployed a linear regression model trained with the original dataset (refer to model specifics in the updated Methods section), which yielded a model for predicting ToD_{MR} values based on the cellular factors. Subsequently, we conducted new experiments on five previously untested cell lines and applied five selected drugs. These experiments encompassed new time-of-day experiments, new long-term luciferase circadian, and new growth, and drug-response live imaging recordings across a spectrum of doses (all new datasets incorporated in the new Supplementary Table 3). From these fresh recordings, we proceeded to analyze all data and derived a new set of clock, growth, and drug sensitivity metrics. Finally, we evaluated the predictive performance of the ToD_{MR} model trained on the original data, but now applied to the newly acquired dataset.

To quantitatively evaluate the accuracy of our predictive model, we employed a Bland-Altman approach, comparing predicted versus actual ToD_{MR} values. Bland-Altman plots provide a visual representation of the agreement between the predicted values from a model and the actual observed values. This approach revealed mean biases of -0.04, -0.05, and 0.06 for the clock, growth, and drug sensitivity datasets, respectively (see Fig. 3). Mean biases near zero (solid horizontal line) and the majority of new data points falling within the limits of agreement (dashed horizontal lines) indicates minimal overall bias and good agreement between the predicted and observed data points. We have incorporated this new analysis into Supplementary Figure 5c of our revised manuscript. Furthermore, we discuss these findings on page 13, lines 379-387.

This analysis underscores the potential of our approach in predicting ToD sensitivity. However, it is essential to acknowledge the inherent limitation of the small sample size, which warrants caution in interpretation.

Furthermore, we have revisited our manuscript and replaced instances of the term "predictive" with phrases such as "highly contributing" or "most important" factors to describe factors exhibiting strong associations with ToD sensitivity. We believe these adjustments more accurately convey the focus of our study. Notably, these changes are reflected on pages 13, lines 392-402, in the revised version of the manuscript.

Figure 3 | Bland-Altman plots comparing predicted and actual ToD_{MR} values of up to 5 new cell lines per drug. The predictions were based on fitting a linear regression model to the original data associated with Pearson correlation coefficients ≥ 0.5 (refer to Fig 5d-f for respective correlation coefficient matrices). This resulted in 26, 8, and 24 new datapoints for the clock, growth and drug sensitivity datasets, respectively. The individual parameters considered

for each dataset are distinguished by markers. The tested drugs are uniformly color-coded across all three datasets. The central solid line indicates the mean bias between the measured and predicted new ToD_{MR} values and the outer lines ($\pm 1.96 \times \text{s.d.}$) mark the upper and lower limits of agreement.

3. Throughout the manuscript, all the replicates are described as “technical”, and there are no mentions of biological replicates. While this might just be a matter of defining “technical” vs. “biological” replicates when using cell lines, can the authors please specify if the replicates were run on different days? In my opinion, independent end-to-end replicate experiments run on different days with the same cell line are sufficient to qualify as biological in this context.

We are sorry that in the original version of our manuscript this was not clearly defined. Bioluminescence data was acquired from two biological replicates (2 different days), using two to three technical replicates per reporter cell line, tested on independent dishes. Experiments involving long-term live-cell microscopy were conducted on the same day from two technical replicates (two separate plates) acquiring images from two different imaging positions within each well. This approach was implemented for screening purposes and based on the capability of long-term live imaging to directly capture and quantify cell growth. We have now more clearly described how our data was acquired and included this in the revised Methods section on page 20, lines 616-619 and lines 631-633. Corresponding changes have also been made to the figure legends to accurately reflect our replication approach.

Minor concerns:

1. Fig. 1 does not contain information about experimental results and is highly redundant with other panels of other figures. As such, it seems more fit for a graphical abstract than as a manuscript figure. In addition, I seldom observe figures being mentioned in the introduction section of a manuscript. I would remove Fig. 1 from the manuscript.

We thank the reviewer for their feedback regarding Figure 1 and the suggestions regarding its placement within the manuscript. We acknowledge the potential redundancy of this figure with other panels and its resemblance to a graphical abstract. However, considering the potentially diverse background of the readership and the interdisciplinary nature of our study, we believe that including this overview early in the text serves to provide valuable context and facilitate understanding of the various methodologies employed. Therefore, we respectfully propose to retain Figure 1 in the manuscript.

2. All throughout the manuscript, when p -values are reported, please specify which comparisons/tests were taken in consideration for each statistical test. Also, I suggest the authors consult a statistician on whether running multiple pairwise t -tests when having multiple comparisons, rather than an ANOVA test with a post-hoc test, is the appropriate approach for testing statistical significance with more than two experimental conditions.

We thank the reviewer for highlighting the need for precise p -value reporting throughout our manuscript. We have now revised our manuscript to clarify the specific statistical tests associated with each calculated p -value.

In addition, as suggested by the reviewer, we now changed our statistical evaluation approach to using an ANOVA test with a post-hoc test and adapted the figure legend to Figure 2 accordingly as well as the corresponding Methods section “Statistical analysis” on page 23, lines 703-706. The implementation of ANOVA over multiple pairwise t -tests is indeed more statistically sound for comparing multiple groups (in our case U-2 OS WT, *Cry1*-sKO, and *Cry1/2*-dKO cells) and mitigates the increased risk of Type I error. We have corrected Figure 2 to present the ANOVA results, and the corresponding p -values have been updated in the results section on pages 5 and 6.

Furthermore, during the re-evaluation of our statistical approach, we identified an error in the previous version of the legend for Figure 3n, Figure 5d, and Supplementary Figure 5a, where a t -

test was incorrectly mentioned. The p -values presented actually represent the statistical significance underlying the correlation coefficients for the various combinations examined. This has now been rectified in the respective figure legends.

3. The Github repo link provided in the manuscript gives me a 404 error when trying to navigate that webpage. Please verify that the link is functional. It would really benefit the community if the code base used to generate the results presented in this manuscript could be reutilized for future projects.

We apologize for the inconvenience caused by the 404 error. The repository was set to private as we finalized the code and prepared additional documentation to ensure that it is user-friendly and robust for public use. We are happy to share that we now made the repository <https://github.com/Granada-Lab/Time-of-Day-Drug-Response> publicly accessible.

4. While the authors provide a link to the code base for the analysis (See point 5 above), in the data availability section they mention that “The experimental time series data and data tables for all results of this study are available upon request” in the Data Availability section. The manuscript is highly computational in nature, and a lot of the results presented here depend on fairly sophisticated analytical pipelines. In the interest of promoting FAIR principles (Which are often required by funders and journals nowadays), I recommend the authors deposit the primary data underlying their manuscript in a generalist repository, such as Zotero, Figshare, or Biostudies.

We thank the reviewer for raising this important concern and we now deposited our experimental data and data tables into the Figshare repository <https://figshare.com/projects/Time-of-Day-Drug-Response/180916>. This will ensure that all our datasets are not only fully accessible post-publication but are also preserved in a stable and searchable environment. Accordingly, we updated the ‘Code availability’ statement on page 27, lines 808-809.

5. Fig. 2D: How was the 95.4% CI calculated? Optional: While all confidence intervals are arbitrary, is there a particular reason why 95.4% was chosen as a threshold, as opposed to the more common 95% value for CIs?

We appreciate the reviewer’s attention to detail regarding the confidence interval (CI) stated in Figure 2d. Our analysis to calculate the CI was based on the ‘autocorr’ MATLAB function. In the documentation of this function, the CI are described as 95.4%. While we initially adopted this, we now realize the unnecessary confusion this number causes. Therefore, we have revised our manuscript to report a standard 95% CI, aligning with common statistical practices and clarifying the documentation’s reference to “approximately 95%” for general applications.

6. Page 5, Line 130: “Heterogeneity between and within cancer entities was further observed for the oscillation period [...]”. What do the authors mean with “cancer entities”? Is it cancer type of origin? Or is it cancer cell line? Please specify.

We apologize for the confusion the term “cancer entities” caused regarding our utilized cancer cell line models. We wanted to highlight the different cancer *types* our various cell lines cover and have now changed the wording from cancer entities to cancer types in the revised version of this manuscript (page 5, now line 131).

7. Page 6, Line 177; and Fig. 3D “To showcase our approach's ability in detecting within-tissue differences, we examined nine cell lines of the triple-negative breast cancer (TNBC) subtype alongside the non-malignant MCF10A breast cell model.” What do the different groups (BL1, BL2, MES) represent? Different stages of breast cancer progression of the tissue from which the cell lines were generated? Please specify and briefly describe in the main text and not only in the abbreviations.

We thank the reviewer for bringing this issue to our attention and apologize for the incomplete description of these abbreviations. BL1, BL2, and MES refer to subtypes of TNBC, as classified by Lehmann *et al.* in 2016. We now briefly describe these abbreviations in the main text (page 7,

lines 192-194) and further mention them in the figure legend to Figure 3 – the first figure where they appear. We also added the abbreviation of EP, short for “epithelial”, to the abbreviation list of the revised manuscript (page 1, line 40).

8. Fig. 3E: What does the shaded area represent? Please specify in the legend. Assuming it is a representation of the error or a CI of the mean of multiple measurements (i.e. the dots in the plot represent a mean of multiple measurements), please also specify the number of replicates used to calculate the mean and the interval.

We thank the reviewer for highlighting this missing information on the shaded area and number of measurements for the data shown in Figure 3e. The datapoints represent the mean normalized cell numbers over time, calculated from two independent assay plates, imaged on the same day. The shaded area around the mean indicates the corresponding standard deviation. We have revised the figure legend to 3d to accurately specify the depicted data.

9. Fig. 3, Legend: “Data in c–f represents the mean \pm s.e.m. from 9 snapshots. k-value errors reflect the 95% CI.” What does the term “snapshot” represent here? Are these 9 independent experiments run on different days? Are 9 cell dishes run in parallel on the same day? If either of these guesses are correct, please use “replicates” and define if these are technical or biological (cfr. point 3).

We regret that the term "snapshots" in the context of Figure 3 caused confusion and we now better clarify this. We have accordingly revised the legend of Figure 3 and removed the word "snapshots" when referring to subfigures c–f.

10. Fig. 4A: I believe that in the right panel the color scheme of the arrows after “sorting” is incorrect, based on the length of the treatment. The red arrow after “sorting” should be the shortest (24 hrs, or 32 – 8 hrs, based on the experimental time above), then blue (28 hrs, or 32 – 4 hrs, based on the experimental time above), then green (32 hrs, or 32 – 0 hrs, based on the experimental time above), then the others, which are already in the correct order. Please double check.

We are grateful to the reviewer for this important observation regarding the color scheme of the arrows in the right panel of Figure 4a. Upon reviewing the figure and the experimental timeline it represents, we can confirm that the red and green arrows were indeed incorrectly depicted in our initial submission. As described by the reviewer, the red arrow should represent the shortest elapsed time to treatment after reset (24 hours, “ToD-0h”), followed by the blue arrow (28 hours; “ToD-4h”), and then the green arrow (32 hours, “ToD-8h”). We have now corrected the color-coding of the arrows to match the corresponding treatment times in the revised version of Figure 4.

11. Page 13, Line 393: “Importantly, the contribution of each gene in discriminating between ToD MR responses varied for the different drugs, suggesting that each drug interacts differently with the molecular components of the circadian clock (Supplementary Fig. 6b–h).”. An alternative explanation for this observation is that expression levels of the target(s) of the drug exhibit circadian variation. There is no evidence in the manuscript that the anti-cancer drugs used here (Which have substantially different cellular targets) interact directly with circadian clock regulators. To rule out that circadian oscillation of expression levels of the targets underlies the variability of ToD among different drugs, the authors should perform qRT-PCR quantification of mRNA expression and/or western blots for a panel of these targets at different clock times.

We thank the reviewer for this insightful comment about our observation that circadian clock genes contributed differently to our the ToD_{MR} values.

Our manuscript indeed does not prove direct interaction between anti-cancer drugs and circadian clock regulators, and we apologize for the potentially confusing statements in the previous version of our manuscript. We now removed that text passage and elaborate more on the potential

explanation mentioned by the reviewer, adapting the manuscript on page 17, lines 522-528 accordingly.

The suggestion to conduct a qRT-PCR experiment to assess mRNA expression of certain drug targets is very interesting and holds promise for revealing insights into the mechanisms of time-of-day drug sensitivity. However, delving into the mechanistic underpinnings of these responses exceeds the scope of our current study. We have now incorporated a discussion point on these proposed experiments into our manuscript's Discussion section to suggest compelling avenues for future research, which could center on the important matter of elucidating the molecular mechanisms driving time-of-day sensitivity.

12. Page 14, Line 437” “Notably, in agreement with recent studies³⁹, our results indicate that the sensitivity metrics exhibit milder correlations, challenging the conventional binary classification of sensitive versus resistant models and suggesting that drug sensitivity rankings are rather metric-specific and drug dependent (Fig. 3).” Please specify which “sensitivity metrics” you are referring to (12a). Also, “milder correlations” compared to which treatment/conditions/etc? (12b)

We are sorry for the confusion generated in the previous version of the manuscript and appreciate the opportunity to clarify these aspects.

- a. With “sensitivity metrics” we refer to the growth inhibition (*GR*) dose-response metrics that we utilized for the assessment of a cell line’s drug sensitivity in Fig 3. Specifically, we are referring to the GR_{50} , GEC_{50} , GR_{AOC} , GR_{inf} , and Hill coefficient values, each offering a different perspective on a cell line’s drug sensitivity, resulting in a comprehensive understanding of drug response dynamics. We now more explicitly describe them (see below).
- b. With “milder correlations” we refer to the cross-correlations of the different drug sensitivity metrics mentioned above and which we show in Figure 3n of our manuscript. The fact that these metrics do not generally correlate with each other agrees with the observations by Fallahi-Sichani *et al.* (reference number 39 in the manuscript), which reported that correlations vary based on the drug class and cell model under investigation.

To improve the clarity and to add more context to the statements made, we have now incorporated more detailed information on page 16, lines 485-488 of the revised manuscript.

****Reviewer-independent changes to the manuscript****

1. In the Material and Methods section on pages 22+23, lines 577 and 682, we corrected and expanded our description of the utilization of either an adaptable or fixed ridge threshold. In the original version of the manuscript, we mistakenly referred to the adaptable threshold for the “ridge length” metric. In fact, we used the adaptable threshold for the extraction of the “period” and “phase difference”, while we applied the fixed threshold for the clock-strength related “amplitude” and “ridge length” metrics.
2. In the Material and Methods section on page 23, lines 680-681 we included an additional tool (‘Circular Statistics Toolbox’ v1.21.9.0. by Philipp Behrens) that was used to calculate the phase difference between *Bmal1* and *Per2* signals shown in Supplementary Fig. 1b of our manuscript.
3. In the Material and Methods section on page 27 line 789 the word “reduction” for “Supervised dimensionality reduction” was missing.
4. In Supplementary Figure 1b we noticed a mistake in the polar histogram axis. The point between $5\pi/4$ and $7\pi/4$ is “ $3\pi/2$ ” and not “ $3\pi/4$ ”.

References

- 1 Börding, T., Abdo, A. N., Maier, B., Gabriel, C. & Kramer, A. Generation of Human CRY1 and CRY2 Knockout Cells Using Duplex CRISPR/Cas9 Technology. *Front Physiol* **10**, 577 (2019). <https://doi.org/10.3389/fphys.2019.00577>
- 2 Lin, H.-H., Qraitem, M., Lian, Y., Taylor, S. R. & Farkas, M. E. Analyses of BMAL1 and PER2 Oscillations in a Model of Breast Cancer Progression Reveal Changes With Malignancy. *Integr Cancer Ther* **18**, 1534735419836494-1534735419836494 (2019). <https://doi.org/10.1177/1534735419836494>
- 3 Lellupitiyage Don, S. S. *et al.* Circadian oscillations persist in low malignancy breast cancer cells. *Cell Cycle* **18**, 2447-2453 (2019). <https://doi.org/10.1080/15384101.2019.1648957>
- 4 Baggs, J. E. *et al.* Network Features of the Mammalian Circadian Clock. *PLOS Biology* **7**, e1000052 (2009). <https://doi.org/10.1371/journal.pbio.1000052>
- 5 Sothorn, R. B., Lévi, F., Haus, E., Halberg, F. & Hrushesky, W. J. M. Control of a Murine Plasmacytoma With Doxorubicin-Cisplatin: Dependence on Circadian Stage of Treatment. *JNCI: Journal of the National Cancer Institute* **81**, 135-145 (1989). <https://doi.org/10.1093/jnci/81.2.135>
- 6 Wood, P. A., Du-Quiton, J., You, S. & Hrushesky, W. J. M. Circadian clock coordinates cancer cell cycle progression, thymidylate synthase, and 5-fluorouracil therapeutic index. *Molecular Cancer Therapeutics* **5**, 2023-2033 (2006). <https://doi.org/10.1158/1535-7163.Mct-06-0177>
- 7 Lévi, F. A. *et al.* Chronomodulated Versus Fixed-Infusion—Rate Delivery of Ambulatory Chemotherapy With Oxaliplatin, Fluorouracil, and Folinic Acid (Leucovorin) in Patients With Colorectal Cancer Metastases: a Randomized Multi-institutional Trial. *JNCI: Journal of the National Cancer Institute* **86**, 1608-1617 (1994). <https://doi.org/10.1093/jnci/86.21.1608>

Reviewers' Comments:

Reviewer #1:

Remarks to the Author:

The Authors have very carefully and thoughtfully responded to my comments and the comments of other reviewers. The manuscript is now much improved, including having some new mechanistic findings, and is much easier and more straightforward to read. I remain very enthusiastic about the potential impact of this work and have no further concerns.

Reviewer #2:

Remarks to the Author:

The authors of the manuscript have addressed all my concerns ,except for Minor concern #1, in the revisions. I am satisfied with the revised manuscript.

We would like to thank the reviewers again for their work and valuable input throughout the review process of this manuscript. We are happy that the reviewers are satisfied with our revisions and have no further concerns.

****REVIEWER #1** (Remarks to the Author):**

The Authors have very carefully and thoughtfully responded to my comments and the comments of other reviewers. The manuscript is now much improved, including having some new mechanistic findings, and is much easier and more straightforward to read. I remain very enthusiastic about the potential impact of this work and have no further concerns.

****REVIEWER #2** (Remarks to the Author):**

The authors of the manuscript have addressed all my concerns ,except for Minor concern #1, in the revisions. I am satisfied with the revised manuscript.